# Brief communication: Identification of tundra topsoil frozen/thawed state from SMAP and GCOM-W1 radiometers measurements using the spectral gradient method

Konstantin Muzalevskiy[1], Zdenek Ruzicka[1], Alexandre Roy[2], Michael Loranty[3], Alexander Vasiliev[4]

[1]Laboratory of Radiophysics of Remote Sensing, Kirensky Institute of Physics Federal Research Center KSC Siberian Branch Russian Academy of Sciences, Siberian Federal University, Krasnoyarsk, Russia
[2]Département des Sciences de l'Environnement, Université du Québec à Trois-Rivières (UQTR), Trois-Rivières, Centre d'étude Nordique, Québec, Canada
[3]Department of Geography, Colgate University, Hamilton, NY, USA
[4]Laboratory for Cartographic Modeling and Forecasting the State of Permafrost Geosystems, Earth Cryosphere Institute, Tyumen Scientific Centre SB RAS, Russia

*Correspondence to*: Konstantin Muzalevskiy (rsdkm@ksc.krasn.ru)

**Abstract.** From 2015 to 2020, using the spectral gradient radiometric method, the possibility of frozen/thawed (FT) state identification of tundra soil was investigated based on SMAP and GCOM-W1 satellite observations of ten test sites located in the Arctic regions of Canada, Finland, Russia, and the U.S.. It is shown that the spectral gradients of brightness temperature and reflectivity, measured on the frequency range from 1.4 GHz to 36.5 GHz on horizontal polarization with a determination coefficient from 0.775 to 0.834, root-mean-square-error from 6.6 days to 10.7 days, and bias from -3.4 days to +6.5 days, make it possible to identify FT state of the tundra topsoil. Spectral gradient method has a significantly higher accuracy for identification of FT state of tundra soils in relation to single-frequency methods based on the calculation of polarization index.

## 1 Introduction

Microwave radiometry is a promising all-weather method for remote sensing of seasonal soil thawing and freezing cycles in the Arctic region. The microwave emission changes significantly at the phase transitions of water in wet soil, that makes it possible to identify the frozen/thawed (FT) state of the soil. Recently, both single- and multifrequency radiometric methods for the identifying of FT soil state have been proposed. In single-frequency methods implemented in the algorithms of Soil Moisture and Ocean Salinity (SMOS) (Rautiainen et al., 2016, 2014; Roy et al., 2015) and Soil Moisture Active Passive (SMAP) (Derksen et al., 2017; Dunbar et al., 2016) satellites, the polarization index $PR = \frac{1}{2}\frac{Tb_V(1.4)-Tb_H(1.4)}{Tb_V(1.4)+Tb_H(1.4)}$ is used as an indicator of FT soil state. Here $Tb_H(1.4)$ and $Tb_V(1.4)$ are the brightness temperatures measured at horizontal (H) and vertical (V) polarizations, on a frequency of 1.4 GHz (at a viewing angle of ~40°). The time series of the polarization index PR are normalized to average seasonal maximum (winter) and minimum (summer) PR values, which are determined for specific landscape types (Rautiainen et al., 2014). The decision on FT state of the soil is made when the normalized PR passes through

0. Hitherto, a space product for identifying FT soil state based on multifrequency radiometric measurements has not been created.

However, the possibilities of identifying FT soil state have been studied using the polarimetric data of multifrequency radiometers, such as Scanning Multichannel Microwave Radiometer (SMMR), Special Sensor Microwave/Imager (SSM/I), Advanced Microwave Scanning Radiometer for EOS (AMSR-E), AMSR-2 (at a viewing angle of ~50°-55°). Multifrequency methods for the identifying of FT soil state are based on the assessment of effective temperature, emissivity (Zhao et al., 2011, 2017; Hu et al., 2017, 2019) or reflectivity (Muzalevskiy and Ruzicka, 2020, 2021), as well as the spectral gradient of brightness temperature ($\Delta Tb/\Delta f$) in a wide frequency range from f=10.7 GHz to f=37 GHz (Zuerndorfer et al., 1989, 1990, 1992). Research (Zuerndorfer et al., 1989, 1990, 1992) shows that the correlation diagram between $\Delta Tb/\Delta f$ in the frequency range 10.7 GHz - 37 GHz and the brightness temperature Tb(37) measured on a frequency of 37 GHz is a good indicator of FT soil state (testing mainly for the Great Plains area U.S.). In this method, the brightness temperatures, measured by SSM/I (SMMR), were averaged between horizontal and vertical polarizations. Results showed that low values of Tb(37) and negative values of $\Delta Tb/\Delta f$ are an effective criterion for assessing the frozen state of topsoil. In a study by (Zhao et al., 2011) of agricultural areas in the Haihe River valley, China, ground radiometric measurements showed, that identification of FT soil state using the effective emissivity had higher confidence compared to the spectral gradient of radio brightness temperature. In this case, the effective emissivity was calculated as the ratio of brightness temperatures $Tb_H(18.7)/Tb_V(36.5)$, measured on a frequency of 18.7 GHz in the horizontal polarization (H) and on a frequency of 36.5 GHz in the vertical (V) polarization. The algorithm created for identifying FT soil state (Zhao et al., 2011, 2017; Hu et al., 2017, 2019) was implemented on the basis of Fisher's discriminant analysis (Fisher 1936), where the discriminant function was a linear combination of two attributes: effective emissivity $Tb_H(18.7)/Tb_V(36.5)$ and brightness temperature $Tb_V(36.5)$. In the article by (Muzalevskiy and Ruzicka, 2020), the $MPR = \frac{1}{2}\frac{\Gamma_H(1.4)+\Gamma_V(1.4)}{\Gamma_H(1.4)-\Gamma_V(1.4)}$ polarization index was proposed as an indicator of topsoil FT state, where $\Gamma_H(1.4)$ and $\Gamma_V(1.4)$ are the effective reflectivity of topsoil at horizontal and vertical polarizations, estimated on a frequency of 1.4 GHz. In contrast to the effective emissivity in the methods (Zhao et al., 2011, 2017; Hu et al., 2017, 2019), the ratio of reflectivities in the MPR index minimizes the effect of soil roughness and canopy optical thickness. Effective reflectivities $\Gamma_{H,V}(1.4) \equiv 1 - \frac{Tb_{H,V}(1.4)}{Tb_V(6.9)}$ were estimated based on brightness temperatures, measured by SMAP on a frequency of 1.4 GHz and GCOM-W1 at a vertical polarization and a frequency of 6.9 GHz (as an estimation of the effective temperature of topsoil (Muzalevskiy et al., 2016)). A threshold level ~1.0-1.2 in the proposed MPR index, showed a significant correlation with the transitions of the topsoil temperature through 0°C at twelve test sites in the North Slope of Alaska, U.S., the North of Canada, Finland and Russia (Muzalevskiy and Ruzicka, 2020; Muzalevskiy et al., 2021). The proposed method is about 10 days more accurate (Muzalevskiy et al., 2021) to determine FT state in relation to the standard SMAP-SPL3FTP_E product (Dunbar et al., 2016), when compared with weather station data. Until now, the possibility of reflectivity spectral gradients as indicator of FT soil state has not been studied. In addition, the advantages of wide frequency range brightness temperature spectral gradient, in the L-band is as an indicator of topsoil FT state have not been studied. The development of multifrequency

algorithms for classifying topsoil FT state is topical, taking into account that the Copernicus Imaging Microwave Radiometer (Kilic et al. 2018), equipped with a high spatial resolution multispectral polarimetric radiometer 1.4-36.5 GHz (55-5 km) as is expected to be launched in 2028.

## 2 Test sites, ground truth and satellites data

Ten test sites equipped with soil-climatic weather stations and located in the Northern regions of U.S., Canada, Finland, and

Russia were selected to investigate spectral gradient methods for identifying FT state of tundra and boreal forest topsoil. The coordinates of the test sites and their landscape classification are summarized in Table 1 The landscape classification of the test sites is based on a database from (ESA, 2017). The statistics are given for the averaged pixel footprint area (44 km x 44 km), the centers of which coincided with the coordinates of the weather stations at the test sites.

**Table 1** Characteristics of test sites.

| Test sites | Region | Longitude/Latitude | Land cover types (%) | Period of observation |
|---|---|---|---|---|
| Franklin Bluffs (FB) | | 148.7208°W/69.6741°N | e: 81, b: 6, g: 6, f: 4, d: 2 | 24.08.16-30.31.20 |
| SagMAT/MNT (SG) | | 148.6739°W/69.4330°N | e: 71, b: 20, g: 3, f: 3, d: 3 | 01.04.15-22.08.18 |
| Happy Valley (HV) | U.S. | 148.8483°W/69.1466°N | e: 57, b: 38, d: 2, g: 1, f: 1 | 01.04.15-30.31.20 |
| Imnaviat (IM) | | 149.3523°W/68.6397°N | e: 78, b: 16, f: 2, d: 2, g: 2 | |
| Banks Island (BI) | Canada | 119.5615°W/73.2200°N | e: 97, g: 3 | 01.04.15-30.06.19 |
| Lake Chisapaw (KJ) | | 76.3141°W/54.9731°N | a: 60, d: 20, g: 10, e: 9 | 30.08.16-19.05.19 |
| Sodankylä (SO) | Finland | 26.6333°E/67.3621°N | a: 85, c: 12, g: 3 | 01.04.15-31.05.19 |
| Saariselka (SA) | | 27.5506°E/68.3302°N | a: 70, c: 23, e: 5, b: 1 | |
| Maresale (MS) | Russia | 66.8100°E/69.7100°N | e: 69, g: 11, b: 12, d: 5, a: 3 | 18.08.15-18.05.19 |
| Cherski (CH) | | 161.4819°E/68.7475°N | a: 53, c: 20, g: 19, f: 5 | 01.04.15-31.03.18 |

a – Forest; b – Grassland; c –Wetland; d – Shrubland; e – Sparse vegetation; f – Bare area; g – Water.

At the North Slope of Alaska test sites soil temperature was measured at a surface (depth of 1cm). FB test site is located in the coastal plain with flat alluvial terrace in a moss tundra landscape with moist nonacidic sedges, and prostrate-shrubs. SG, HV and IM are located on hilly terrain with dominantly moist acidic and non-acidic tussock tundra to the north and considerable shrub growth to the south. The BI test site is formed by gentle hills with sparse vegetation in the form of mosses and grassy

meadows. Boreal forest test sites are represented by SO, SA in Finland and KJ in Canada. SO and SA are located in different density forest, non-forest area is covered with juniper, heather and thin layer of lichen and moss. KJ are located in typical Canadian taiga with low-density black spruce-lichen woodland (sandy soils). Soil temperature at the test sites SO, SA and KJ were measured at a depth of 2-5cm. MS test sites located on the Yamal peninsula in moist and dry dwarf shrub-moss-lichen tundra in combination with sedge-moss mires. The soil temperature at MS was measured at a depth of 2-5 cm. The CH area is

dominated by wetlands covered with larch forests of varying canopy cover (ranging from 13% to 75%). At CH test site soil temperature was measured in five test plots (which averaged for further analysis) at the interface between the organic and mineral soil layers, at depths of 6-10 cm below the surface depending on organic layer thickness (Loranty and Alexander, 2021). At the KJ, MS and CH test sites water objects are occupied more than 10% in the pixel area.

The brightness temperatures of test sites were measured on vertical and horizontal polarizations by SMAP and GCOM-W1 satellites on frequencies of 1.4 GHz and 6.9 GHz, 10.7 GHz, 18.7 GHz, 36.5 GHz, respectively. Ascending SMAP and GCOM-W1 orbits were chosen. SMAP polarimetric brightness temperature data (SPL3FTP_E) in Northern Hemisphere azimuthal projections on a 9 km Equal-Area Scalable Earth Grid (EASE-Grid 2.0) were used over the test sites in the period of soil temperature observations by the weather stations (see Table 1). GCOM-W1 brightness temperature data were acquired from L1R product, where the brightness temperatures of high-resolution channels are resampled to the effective pixel area of lower resolution channel on a frequency of 6.9GHz, gridded with 12.5 km cell size. Pixels closest to the coordinates of weather stations, with the exception of MS, were used in the analysis. MS is located at the coast of the Kara Sea, thus a pixel whose center is far more than 50 km from the sea was chosen. Daily 9 km EASE-Grid freeze/thaw SMAP product (SPL3FTP_E, both ascending and descending orbits) based on the polarization index (Dunbar et al. 2016) was used for identification FT state of land at the test sites.

## 3 Spectral gradient methods for identifying the frozen/thawed state of topsoil

The spectral components of brightness temperature are formed by the emitting layers of a ground half-space with different thicknesses. For this reason, the difference between brightness temperatures measured in the different parts of frequency spectrum is related to the vertical (in the direction of the 0z axis) gradient $\frac{\Delta T_g}{\Delta z}\Big|_{z=0}$ of ground temperature $T_g$, and hence with the direction of the heat flux $J_0 = K\frac{\Delta T_g}{\Delta z}\Big|_{z=0}$, where K is the thermal conductivity, through the soil surface. Indeed (Zuerndorfer and England, 1992), on the basis of the phenomenological theory of emission, the brightness temperature $Tb_p(f)$ of a dielectric half-space with a linear temperature profile can be represented as

$$Tb_p(f) = [1 - \Gamma_p(f)]T_{g0} + [1 - \Gamma_p(f)]\frac{\Delta T_g}{\Delta z}\Big|_{z=0} z_{eff}(f), \tag{1}$$

where f is the frequency electromagnetic field, $\Gamma_p(f)$ is the ground reflectivity in vertical $p$=V or horizontal $p$=H polarization, $T_{g0}$ is the temperature of ground surface, $z_{eff}(f)$ is the thickness of emitting layer. From equation (1) it follows, that the density of spectral gradient of brightness temperature is directly proportional to the heat flux through the boundary of the ground surface:

$$\frac{\Delta Tb_p(f)}{\Delta f} = -T_{g0}\frac{\Delta\Gamma_p(f)}{\Delta f} + \frac{1}{K}J_0\frac{\Delta(1-\Gamma_p(f))z_{eff}(f)}{\Delta f}. \tag{2}$$

By analogy with the density of spectral gradient of brightness temperature (2), the density of spectral gradient of reflectivity can be introduced $\Delta\Gamma_p(f)/\Delta f$.

For dielectric-inhomogeneous and nonisothermal half-space, the reflectivity at frequency f, in accordance with the method (Muzalevskiy et al., 2021; Muzalevskiy and Ruzicka, 2020), can be estimated based equation:

$$\Gamma_p(f) = 1 - Tb_p(f)/T_{eff}. \tag{3}$$

In equation (3), $T_{eff}$ and $\Gamma_p(f)$ can be interpreted, respectively, as effective ground-canopy temperature and reflectivity, which takes into account soil surface roughness and canopy optical thickness using one combined parameter (Fernandez-Moran et al., 2015, see equation (10)). The effective temperature $T_{eff}$ can be estimated based on the measurements of brightness temperature $Tb_V(6.9)$ by the GCOM-W1 satellite on a frequency of 6.9 GHz on vertical polarization, the values of which correlate with the surface temperature of tundra soil (Muzalevskiy et al., 2016). The physical basis for this estimate is the observation angle of AMSR-2/GCOM-W1, which is close to Brewster's angle (55°) and this leads to a decreased impact of reflectivity $\Gamma_V$ on measured brightness temperature $Tb_V(6.9)$. Indeed, reflection coefficient measurements show that as root-mean-square (RMS) heights of soil surface roughness increase from 0.25 cm to 0.93 cm, Brewster's angle decreases from 60° to 57°. (De Roo and Ulaby, 1994, also see Wang et al., 1983, Fig. 2). The roughness of natural tundra soils has much higher values, which change in a wide range from 1.06 cm to 4.28 cm (Watanabe et al., 2012). The presence of vegetation (snow) cover on a rough soil surface leads to blurring and flattening of the V-pol angular dependence of reflectivity (Rodriguez-Alvarez et al., 2011, see Figs. 7, 8) and brightness temperature (Lemmetyinen et al., 2016, see Figs. 5-7; Chang and Shiue, 1980, see Fig. 3-5) in the region of Brewster's angles, due to the interference phenomenon. Within the error measurement of brightness temperature of $\pm1.3$–1.5 K (Piepmeier et al., 2017; Gao et al., 2019), as well as the accuracy of emission models of $\pm4$–5 K (Wigneron et al., 2011), Brewster's angle can be determined within a wide range of about from 45° to 65° (Lemmetyinen et al., 2016, see Fig. 5-7; Chang and Shiue, 1980, see Fig. 3-5). The error measurement of brightness temperature and the accuracy of emission models also makes it possible to neglect the variations of H-pol brightness temperatures of snow(vegetation)-covered soil in a range of observation angles from 40° to 55° (Roy et al., 2018, see Fig. 3; Lemmetyinen et al., 2016, see Fig. 7; Chang and Shiue, 1980, see Fig. 3-5). In this regard, the use of brightness temperatures measured at different angles in equation (3) is approximate. As a result, and as expected from model (3), $Tb_V(6.9)$ becomes mainly directly proportional to $T_{eff}$. For this reason, the reflectivity was further estimated only for horizontal polarization. Further, estimates of $\Gamma_H(f)$ will be considered as the apparent values of reflectivity, since the absolute value of $Tb_V(6.9)$ does not coincide with the actual values of $T_{eff}$ or $T_{g0}$, but is only proportional to them.

## 4 Results and Discussion

The time series of brightness temperatures $Tb_H$, measured by the SMAP and GCOM-W1 satellites, as well as the reflectivity $\Gamma_H$, calculated based on formula (3) are shown for the HV test site in Fig. 1a, 1b. For the other test sites, such dependencies are similar. The time series of $Tb_H$ and $\Gamma_H$ (see Fig. 1a, 1b) has a pronounced seasonal variation with a periodic change in maximum and minimum values. On the dates corresponding to the moment of soil thawing or freezing, the order of brightness temperature and reflectivity values are inverted, along with an increase or decrease in frequency. Several such time periods are marked by rectangles with a dashed line in Fig. 1a, 1b (see the period 2015-2016).

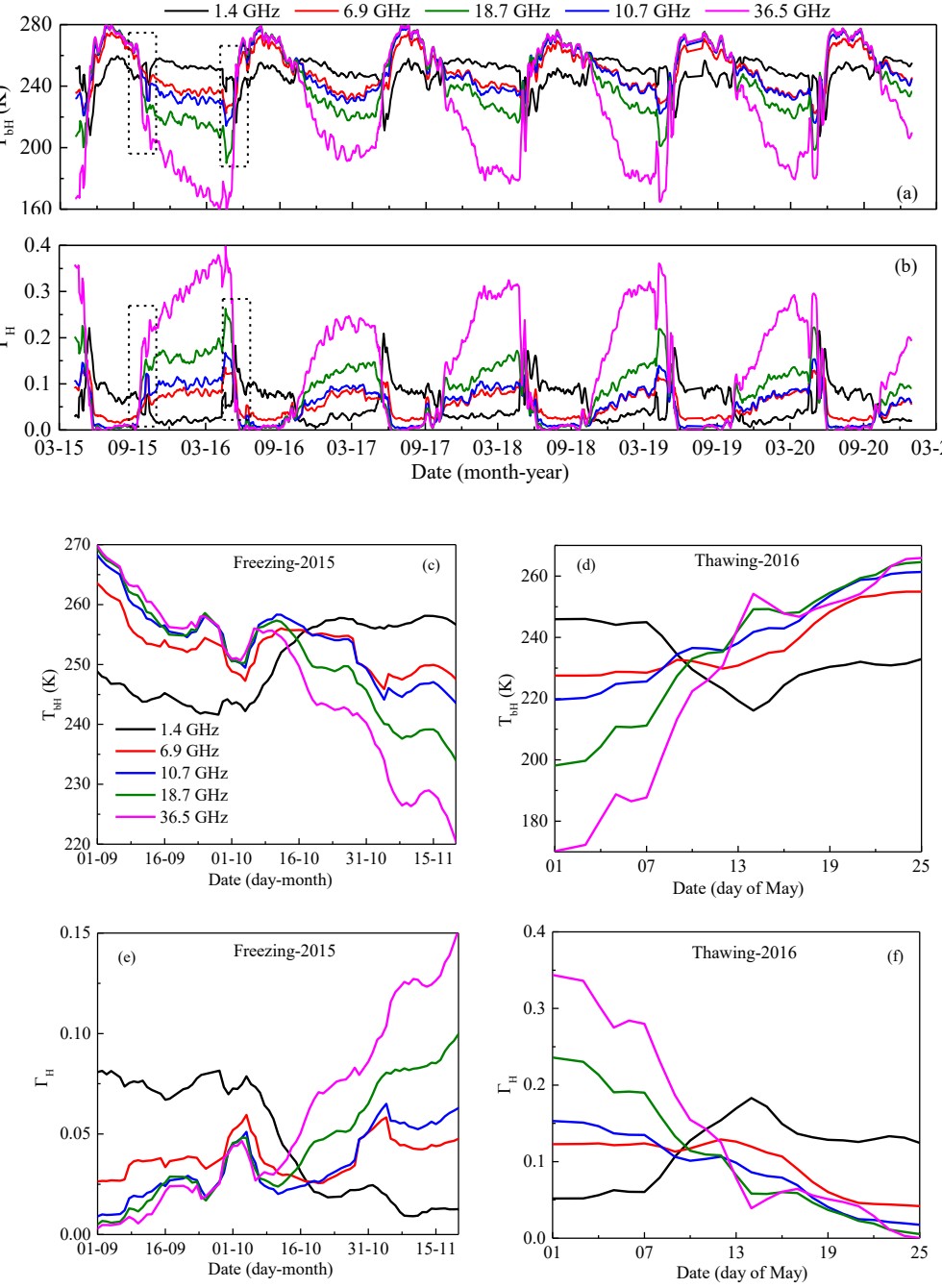

**Figure 1.** Time series of brightness temperatures (a), (c), (d) and reflectivity (b), (e), (f), according to SMAP and GCOM-W1 satellites data in the frequency range from 1.4 to 36.5 GHz for HV test site. (c) and (e) freezing process in 2015, (d) and (f) thawing process in 2016.

In Fig. 1, these areas are shown on a larger scale. Indeed, as can be seen from Fig. 1 in winter (before May 7, 2016 and after November 1, 2015), the values of brightness temperatures decrease with increasing frequency, i.e. a negative gradient is observed, but in summer the positive gradient is observed (see Fig. 1c, 1d). The seasonal variation of reflectivity has an opposite spectral gradient (see Figs. 1e, 1f) with respect to the spectral gradient of brightness temperature. In the transition period (May 7-19, 2016 and October-November, 2015, see Fig. 1), the spectral gradient of brightness temperature and reflectivity is minimal. In these time intervals, for any pairs of $Tb_H(f_1)$ and $Tb_H(f_2)$, $\Gamma_H(f_1)$ and $\Gamma_H(f_2)$ at different frequencies $f_1$ and $f_2$, there is a point of zero spectral gradient (the point of sign change in the spectral gradient), at which the direction of the heat flux must change, in accordance of formula (2). Unlike the methods in which the FT index threshold level is set on the basis of calibration or normalization (Rautiainen et al., 2016; Derksen et al., 2017; Muzalevskiy and Ruzicka, 2020; Muzalevskiy et al., 2021) the zero-threshold level of spectral gradients of brightness temperatures or reflectivity does not require calibration for identification FT topsoil state. FT classification based on the spectral gradient method was applied for the first time by Zuerndorfer et al. (1989).

Detailed depictions of the spectral gradient densities of brightness temperature and reflectivity are depicted in Fig. 2a, 2b for HV test site. To this end, the spectral gradient densities of brightness temperature and reflectivity for frequency pairs: 1.4-6.9 GHz, 1.4-10.7 GHz, 1.4-18.7 GHz and 1.4-36.5 GHz were calculated. The largest variations of brightness temperatures $\Delta Tb_H(f)$ and reflectivites $\Delta\Gamma_H(f)$ with frequency (see Fig. 1) are observed in winter, while the largest variations in the spectral gradient densities of brightness temperature $\Delta Tb_H(f)/\Delta f$ (see Fig. 2a) and reflectivity $\Delta\Gamma_H(f)/\Delta f$ (see Fig. 2b) are observed in summer. The spectral gradient of brightness temperature and reflectivity is highest in the frequency range of 1.4-36.5 GHz, and the lowest in the frequency range of 1.4-6.9 GHz. This is due to a significant contrast of temperatures and permittivities between the shallow and deeper emitting layers of ground at frequencies of 36.5 GHz and 1.4 GHz, respectively, compared to radiation layers that are close in thicknesses on frequencies of 1.4 GHz and 6.9 GHz. At the same time, per unit interval of the frequency spectrum $\Delta f$, the spectral gradient densities of brightness temperature $\Delta Tb_H(f)/\Delta f$ and reflectivity $\Delta\Gamma_H(f)/\Delta f$ seems to be larger for the narrower 1.4-6.98 GHz, than for the wider 1.4-36.5 GHz frequency band. The amplitudes of seasonal variations of $\Delta Tb_H(f)/\Delta f$ are synchronous with variations of surface temperature of soil. The time series of $\Delta\Gamma_H/\Delta f$ varies in antiphase to the soil surface temperature (see Fig. 2b). Figure 2 shows, that the time points corresponding to the transitions of $\Delta Tb_H(f)/\Delta f$ and $\Delta\Gamma_H/\Delta f$ through 0 are well correlated with the time of soil surface temperature $T_{s0}$ transition through 0°C.

Similar patterns in the behavior of $\Delta Tb_H(f)/\Delta f$ and $\Delta\Gamma_H(f)/\Delta$ are also observed for other test sites, except SO and SA. As an example, $\Delta\Gamma_H(f)/\Delta f$ for KJ, CH, SO, and SA test sites are shown in Fig. 2c-2f. In some years, for SO and SA test sites, multiple passing of $\Delta\Gamma_H(f)/\Delta f$ through 0 during winter can be observed (see Fig. 2e, 2f), which does not allow unambiguous identification of FT states. These processes will be explained below. Also, note that the presence of significant wetland and open water at the CH test site is not detected in the behavior of $\Delta\Gamma_H(f)/\Delta f$ ($\Delta Tb_H(f)/\Delta f$) in comparison with other test sites.

For further analysis, seasonal variations of $\Delta Tb_H(f)/\Delta f$ and $\Delta\Gamma_H/\Delta f$ will be used, the amplitudes of which take maximum (see Fig. 2, curve 1) and minimum (see Fig. 2, curve 4) values in the frequency ranges of 1.4-36.5GHz and 1.4-6.9 GHz,

respectively. The soil will be considered frozen or thawed when the soil surface temperature, $T_{s0}$, transits through 0°C, according to weather stations installed at test sites.

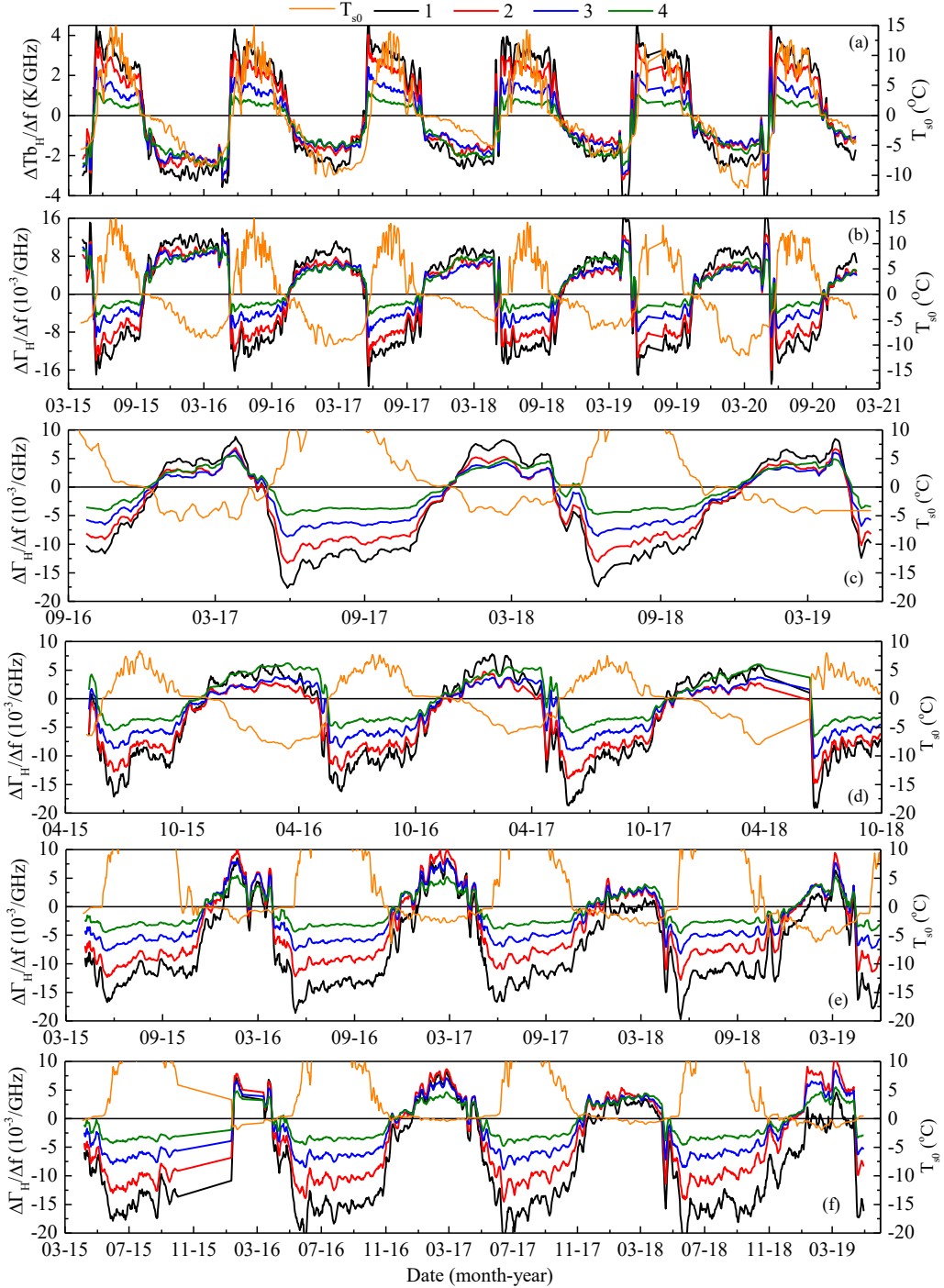

**Figure 2.** Density of spectral gradients of brightness temperature (a) and reflectivity (b) for HV test site, and density of spectral gradients of reflectivity for (c) KJ, (d) CH, (e) SO, (f) SA test sites, calculated for the pairs of frequencies: 1) 1.4-6.9 GHz, 2) 1.4-10.7 GHz, 3) 1.4-18.7 GHz and 4) 1.4-36.5 GHz.

The synchronicity of transition through the zero spectral gradient densities threshold of brightness temperature $\Delta Tb_H/\Delta f$ and reflectivity $\Delta\Gamma_H/\Delta f$ with the soil surface temperature crossing 0°C were estimated (see Fig. 3).

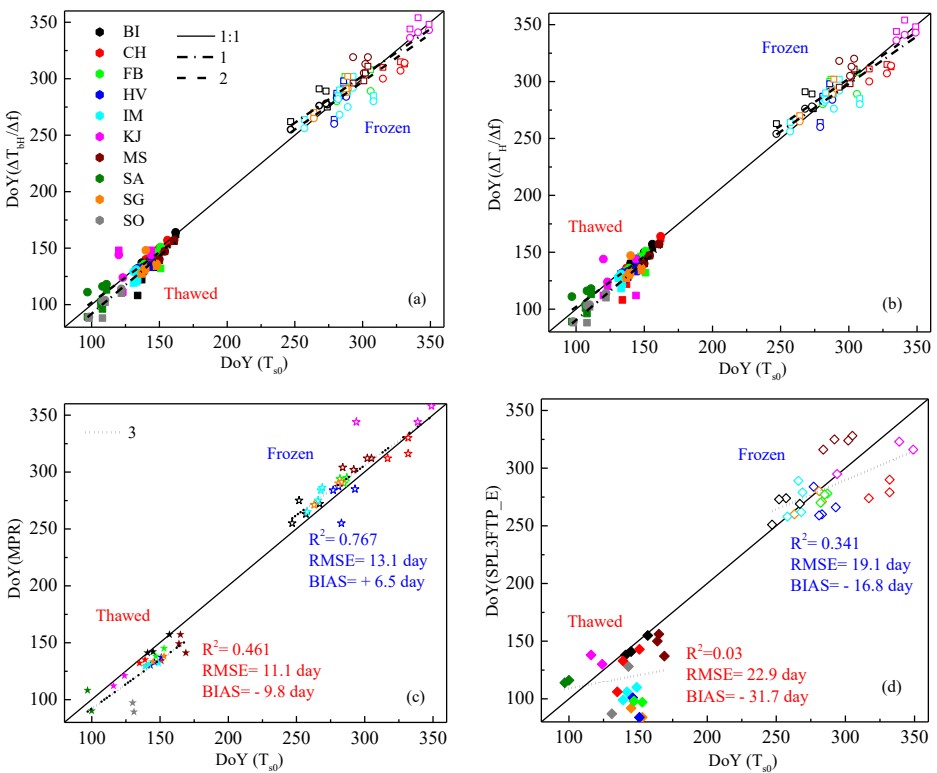

**Figure 3.** Correlation between days of the year (DoY), for which: $\Delta Tb_H/\Delta f$ (a), $\Delta\Gamma_H/\Delta f$ (b), MPR (c) crosses threshold level, (d) soil becomes FT based on SMAP SPL3FTP_E product and soil surface temperature stable crossing 0°C. Different test sites are marked with different colors. The open and filled symbols indicate the frozen and thawed topsoil state, respectively. Squares and circles indicated a spectral range of 6.9-1.4 GHz and 36.5-1.4 GHz, respectively. Regression lines for the spectral ranges 6.9-1.4 GHz and 36.5-1.4 GHz are marked as (1) and (2), respectively. Regression line (3) corresponds to MPR index and SMAP SPL3FTP_E product.

From the correlation analysis of the data presented in Fig. 3, it follows that the $\Delta Tb_H(f)/\Delta f$ and $\Delta\Gamma_H/\Delta f$ in the frequency range

of 6.9-1.4 GHz (36.5-1.4 GHz) determines the thawed state of the topsoil by 8.4-8.9 (1.2-3.3) days earlier, relative to soil surface temperature (see Table 2, systematical error (SE)).

Table 2. Determination coefficient ($R^2$), root-mean-square-error (RMSE), systematic error (SE) in identifying the FT topsoil state.

| | $\Delta Tb_H/\Delta f$ | | | | $\Delta \Gamma_H/\Delta f$ | | | |
| --- | --- | --- | --- | --- | --- | --- | --- | --- |
| | Thawed | | Frozen | | Thawed | | Frozen | |
| $\Delta f$ (GHz) | 6.9-1.4 | 36.5-1.4 | 6.9-1.4 | **36.5-1.4** | 6.9-1.4 | **36.5-1.4** | 6.9-1.4 | 36.5-1.4 |
| $R^2$ | 0.802 | 0.833 | 0.819 | **0.775** | 0.855 | **0.834** | 0.816 | 0.784 |
| RMSE (day) | 8.5 | 6.7 | 9.2 | **10.7** | 7.1 | **6.6** | 9.1 | 10.4 |
| BIAS (day) | -8.4 | -3.3 | +5.7 | **-3.4** | -8.9 | **-1.2** | +5.6 | +6.5 |

In the frequency range of 6.9-1.4 GHz, the frozen topsoil state is determined on average 5.6-5.7 days later (see Table 2, both indexes). In the frequency range of 36.5-1.4 GHz, $\Delta Tb_H/\Delta f$ makes it possible to determine the frozen topsoil state in 3.4 days earlier, and $\Delta \Gamma_H/\Delta f$ in the frequency range of 6.9-1.4 GHz in 6.5 days later (see SE in Table 2). In general, both indexes $\Delta Tb_H/\Delta f$ and $\Delta \Gamma_H/\Delta f$ have similar values of RMSE in predicting FT topsoil state. At the same time for the identification of a

200 thawed topsoil state, $\Delta \Gamma_H/\Delta f$ has a greater accuracy (see Table 2, SE and RMSE is lower, $R^2$ is higher compared to $\Delta Tb_H/\Delta f$) in the spectral range of 36.5-1.4 GHz. $\Delta Tb_H/\Delta f$ is more suitable for identifying a frozen topsoil state in the frequency range of 36.5-1.4 GHz (see Table 2), since it has a smaller SE (at comparable RMSE and $R^2$) with respect to $\Delta \Gamma_H/\Delta f$. Spectral gradient methods have improved accuracy in identifying FT topsoil state in relation to both the MPR index (see Fig. 3c and Table 2, column in bold) and to the standard SMAP product SPL3FTP_E (see Fig. 3d and Table 2, column in bold).

It should be noted at correlation analysis (in the case of frozen state) did not take into account SO and SA test sites, since the indexes of $\Delta Tb_H/\Delta f$ and $\Delta \Gamma_H/\Delta f$ led to a systematic error of about 1.5 months, due to unstable soil freezing (soil surface temperature for most of the winter ranged from 0°C to -2°C-5°C according to weather stations, see Fig. 2f, 2e). This systematic error may be because, in contrast to the other test sites, SO and SA stand out by the largest share of a forest, from 70% to 85% in the footprint (see Table 1). Apparently, such a significant share of forests contributes to the formation of a thicker forest

litter and increased accumulation of snow cover compared to the rest test sites, as well as to the test sites KJ and CH, the share of forests in which is less (60% and 53%, respectively). More thick forest litter and snow cover create additional inertia in the thermal exchange between air and soil. Indeed, according to SO and SA stations data, these test sites have characteristics of small negative soil surface temperatures in winter, as well as the extended period of zero-curtain effect (see Fig. 2e, 2f). As a result, in some years, the unstable-transition FT state of topsoil is reflected in the unstable transition of $\Delta \Gamma_H(f)/\Delta f$ ($\Delta Tb_H/\Delta f$)

through 0 during winter, which explains the observed systematic error on SO and SA test sites. Similar unstable transitions of brightness temperature have been observed in ground-based radiometric experiments at the SO test site (Lemmetyinen et al., 2016, see Fig. 2 "wetland 2013-2014"); thawed soil under dry snow layer is a common phenomenon (Kumawat et al., 2022).

## 5 Conclusion

In this article the spectral gradient methods were used to identify FT topsoil state of northern regions based on brightness
temperatures measurements from SMAP and GCOM-W1 satellites across a wide frequency range. As criteria for determining

FT topsoil state, the spectral gradient densities of brightness temperature and reflectivity were used in the frequency range from 1.4 GHz to 6.9 GHz and from 1.4 GHz to 36.5 GHz. Both criteria give the comparable accuracy of forecasting FT topsoil state for tundra and boreal forest regions. At the same time, to improve the accuracy of thawed topsoil state predictions, the spectral gradient density of reflectivity should be used, but for identification of frozen topsoil state, the spectral gradient density of brightness temperature should be used (the frequency range of 1.4 GHz-36.5 GHz). The proposed method makes it possible to identify forest soils are in a transitional state (the soil surface temperature is about 0°C or has small negative values), which is revealed in the multiple transitions of spectral gradient densities of brightness temperature and reflectivity through 0 (more pronounced in the frequency range of 1.4-6.9 GHz). The methods considered in this work do not fully separate the contributions of FT topsoil state from those of water bodies. The freezing processes of open water areas with the formation of ice, the description of which requires a large number of additional data on snow cover thickness, air and water temperature, and wind speed are particular difficult to account for. We did not find any significant differences in the behavior of the spectral gradient densities of brightness temperature and reflectivity, measured for the test site with a high share of wetland (20%) and open water (19%) from the other test sites. Despite all assumptions made in the proposed method, the identification of FT soil surface states is possible with a relatively high determination coefficient of 0.775-0.834, small root-mean-square-error of 6.6-10.7 days, and bias from -3.4 to +6.5 days. Further validation of this methodology, requires expanding the number of test sites, as current analysis is limited by the small number of soil-climatic weather stations with available and up-to-date soil active layer temperature data.

**Financial support**

This research has been supported by the state assignment of Kirensky Institute of Physics. Weather stations data was support by Canadian Space Agency, NSERC and FRQNT; US National Science Foundation [PLR-1304464 and PLR-1417745]; State research programs [121041600043-4].

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
