# Peer review of "Brief communication: Identification of tundra topsoil frozen/thawed state from SMAP and GCOM-W1 radiometers measurements using the spectral gradient method"

_EGUsphere, 2022_

## Author Comment (AC1)

**Response to CC1: 'Comment on egusphere-2022-224', Vasiliy Tikhonov, 28 Jul 2022.**
**https://doi.org/10.5194/egusphere-2022-224-CC1**

**Comment 0.** Dear Editors,

Two months ago the same article by the same authors, but in the Russian language, was received by Sovremennye Problemy Distantsionnogo Zondirovaniya Zemli iz Kosmosa (http://jr.rse.cosmos.ru/?lang=eng). As a reviewer, I recommended to reject it (see my review below).

After that the article was sent to two journals: Issledovanie Zemli iz Kosmosa (https://sciencejournals.ru/journal/iszem/, http://www.jizk.ru/) in Russian and The Cryosphere (https://www.the-cryosphere.net/) in English.  The Russian journal also rejected the article after a negative review by another reviewer.

I believe the authors violated publication ethics by sending the same article to two journals at the same time.

Kind regards,
Dr. Vasiliy Tikhonov
Senior Scientist
Head of Laboratory for Satellite Monitoring of the Earth's Cryosphere
Department of Earth Research from Space
https://iki.cosmos.ru/research/issledovanie-zemli-iz-kosmosa,
Space Research Institute RAS
https://iki.cosmos.ru/
E-mail: vtikhonov@asp.iki.rssi.ru
vasvlatikh@yandex.com
My review of the article sent to Sovremennye Problemy Distantsionnogo Zondirovaniya Zemli iz Kosmosa  (http://jr.rse.cosmos.ru/?lang=eng)

**Response to comment 0.** The persistence with which Vasiliy Tikhonov (PhD) hunts on our manuscript indicates a deep conflict of interests of Vasiliy Tikhonov (PhD) likely not only with the subject of our work, but with the first author.

The official license agreement filled out by the editors of "Issledovanie Zemli iz Kosmosa", and sent to me was not signed by me on purpose, because I submitted the manuscript to The Cryosphere. For this reason, I did not give legal obligations not to publish the article in other journals and did not give the right to publish our article in the journal "Issledovanie Zemli iz Kosmosa".

The submitted manuscript to The Cryosphere is a continuation of our research. (This is the third article in our research cycle). The Russian version of this article was rejected by the journal "Sovremennye Problemy Distantsionnogo Zondirovaniya Zemli iz Kosmosa" in 2022 based on a single expert decision (without the possibility respond to reviewer comments).  The second article from our series was submitted to the journal  "Sovremennye Problemy Distantsionnogo Zondirovaniya Zemli iz Kosmosa" last year (03/02/2021) and  it also was rejected. However, later, with almost no changes in the main content, the second article in our series was successfully reviewed and published in (Muzalevskiy et al., 2021). The first article in our series was published in (Muzalevskiy and Ruzicka, 2020).

**Reference**

Muzalevskiy K., Ruzicka Z., Roy A., Loranty M., Vasiliev A. Classification of the frozen/thawed surface state of Northern land areas based on SMAP and GCOM-W1 brightness temperature observations at 1.4 GHz and 6.9 GHz. Remote Sensing Letters. 2021. Vol. 12. No. 11. P. 1073-1081. 10.1080/2150704X.2021.1963497

Muzalevskiy K., Ruzicka Z. Detection of soil freeze/thaw states in the Arctic region based on combined SMAP and AMSR-2 radio brightness observations // International Journal of Remote Sensing. – 2020. – V. 41.- Is. 14. – P. 5046-5061. DOI: 10.1080/01431161.2020.1724348.

I would like to have more objective assessment of our work by independent experts, and not only Vasiliy Tikhonov (PhD).

I have responded on behalf of all the authors to all comments Vasiliy Tikhonov (PhD), most of which are written in disrespectful, aggressive and peremptory form. Response to the comments are contained in the attached file.

Best regards,
Muzalevskiy Konstantin (PhD)
Head of Laboratory of Radiophysics of the Earth Remote Sensing,
Kirensky Institute of Physics,
Federal Research Center KSC Siberian Branch RAS

**Comment 1.** The presentation of the material is awful, both the narration and the physics of the problem. There are plenty of trivial or mutually exclusive statements throughout the text. The analysis of the satellite data is done by juggling the values of either brightness temperatures for different channels, or their combinations. The suggested physical interpretations also leave much to be desired.

**Response to comment 1.** These statements are general, non-concrete and can be simply copied without change for reviewing of any other article, devoted to the problems of remote sensing. These statements cannot be answered because they are not concrete. Reasoned responses to all other comments are given below.

Here I only want to give reference to such works as

(Pulliainen et al, 1997),

$$
\begin{aligned}
T = {} & 0.2769 \cdot T_{b,19\mathrm{V}} - 0.5101 \cdot T_{b,19\mathrm{H}} \\
& + 0.9758 \cdot T_{b,22\mathrm{V}} + 0.6959 \cdot T_{b,37\mathrm{V}} \\
& - 0.4244 \cdot T_{b,37\mathrm{H}} - 0.038\,12 \cdot T_{b,85\mathrm{V}} \\
& - 0.027\,16 \cdot T_{b,85\mathrm{H}} - 243.4.
\end{aligned}
\tag{9}
$$

$$
\begin{aligned}
T = {} & 0.2510 \cdot T_{b,19\mathrm{V}} + 0.7492 \cdot T_{b,19\mathrm{H}} \\
& + 0.2093 \cdot T_{b,22\mathrm{V}} - 0.4387 \cdot T_{b,37\mathrm{V}} \\
& + 0.1079 \cdot T_{b,37\mathrm{H}} + 0.1703 \cdot T_{b,85\mathrm{V}} \\
& - 0.0254 \cdot T_{b,85\mathrm{H}} - 254.8.
\end{aligned}
\tag{10}
$$

$$T = 0.8315 \cdot T_{b19V} - 0.9173 \cdot T_{b,19H}$$
$$+ 1.022 \cdot T_{b,22V} - 240.0. \qquad (11)$$

$$T_{\text{phys}} = \frac{kT_{b,19V} - (k-1)T_{b,19H}}{e_x}. \qquad (12)$$

(McFarland et al, 1990)

$$ST = C_1 * TBV \qquad (1)$$

$$ST = C_0 + C_1 * 37\,V - C_2 * 22\,V$$
$$- C_3 * 19\,H + C_4 * 85\,V \qquad (2)$$

$$ST = A_1 * 37\,V + A_2 * (37\,V - 22\,V)$$
$$+ A_3 * (37\,V - 19\,H) + A_4 * 85\,V. \qquad (3)$$

$$ST = C_0 + (A_1 + A_2 + A_3) * 37\,V - A_2 * 22\,V$$
$$- A_3 * 19\,H + A_4 * 85\,V. \qquad (4)$$

$$ST = C_0 + C_1 * 19\,H + C_2 * 22\,V$$
$$+ C_3 * 37\,V + C_4 * 85\,V. \qquad (5)$$

(Jones et al,  2007)

$$T_s = \beta_0 + \beta_1 T_{\text{bv6}} + \beta_2 T_{\text{bv10}} + \beta_3 T_{\text{bv23}} + \beta_4 T_{\text{bv89}} + \beta_5 \zeta_{89}$$
$$(12)$$

Are there "plenty of trivial or mutually exclusive statements", "The analysis of the satellite data is done by juggling the values of either brightness temperatures for different channels, or their combinations", "The suggested physical interpretations also leave much to be desired" in (Pulliainen et al, 1997, McFarland et al, 1990, Jones et al, 2007)?

Apparently, all works in which remote sensing data are  "juggled" using artificial intelligence technique (Santi et al, 2014; Gao et al, 2022) also deserve similar assessments from Vasiliy Tikhonov (PhD)?

**References**

Calvet, J.-C., J.-P. Wigneron, E. Mougin, Y. H. Kerr, and J. S. Brito (1994): Plant water content and temperature of the Amazon forest from satellite microwave radiometry. IEEE Trans. Geosci. Rem. Sens., 32, 397-408.

Gao L, Qiang Gao, Hankui Zhang, Xiaojun Li, Mario Julian Chaubell, Ardeshir Ebtehaj, Lian Shen, Jean-Pierre Wigneron, A deep neural network based SMAP soil moisture product, Remote Sensing of Environment, Volume 277, 2022, 113059.

Jones L. A., J. S. Kimball, K. C. McDonald, S. T. K. Chan, E. G. Njoku and W. C. Oechel, "Satellite Microwave Remote Sensing of Boreal and Arctic Soil Temperatures From AMSR-E," in IEEE Transactions on Geoscience and Remote Sensing, vol. 45, no. 7, pp. 2004-2018, July 2007, doi: 10.1109/TGRS.2007.898436.

McFarland, M. J., R. L. Miller, and C. M. U. Neale (1990): Land surface temperature derived from the SSM/I passive microwave brightness temperatures. IEEE Trans. Geosci. Rem. Sens., 28(5), 839-845.

Pulliainen, J. T., J. Grandell, and M. T. Hallikainen (1997): Retrieval of surface temperature in Boreal forest zone from SSM/I data. IEEE Trans. Geosci. Rem. Sens., 35, 1188-1200.

Santi, E.; Pettinato, S.; Paloscia, S.; Pampaloni, P.; Fontanelli, G.; Crepaz, A.; Valt, M. Monitoring of Alpine snow using satellite radiometers and artificial neural networks. Remote Sensing of Environment. 2014. 10.1016/j.rse.2014.01.012

**Comment 2.** The first thing I would like to point out is the definitions of "thawed" and "frozen" states of soil, which are absent in the article. Abstract reads: "The data of soil-climatic weather stations at key sites on soil surface temperature at the transition through 0°C were used for ground validation of the thawed/frozen state of soil". I would like to stress that this is fundamentally incorrect, because soil can be thawed at negative temperatures as well (all depends on the soil structure) (see, for example, Ulaby, Long, 2014). Well, let's leave it to the conscience of the authors. However, at the very end of Results and Discussion (p. 12), Authors report that two sites (SO and SA) were removed from the analysis due to big errors. Authors attribute these errors to "…unstable soil freezing (soil surface temperature for most of the winter ranged from 0°C to -2°C-4°C according to weather stations)". So, what is the "frozen" state of the soil, if from 0 to -4°C it is "not frozen"?

**Response to comment 2.**

1. Our manuscript contains only 11p.

2. The idea that temperature is not an indicator of thawed or frozen ground is not new. For example, in a number of works, including our, it has been shown that changes in the values (jumps) of permittivity during the phase transitions of soil water are the best indicator of the thawed or frozen soil states. (Mironov and Muzalevskiy, 2014; Roy et al, 2017; Tao et al, 2019).

[Figure]

x-axis caption: Температура почвы, °C

Fig. 2. Temperature dependence of soil permittivity. 1 - Retrieved values from SMOS radiometric data, 2 - Regression dependence, 3 - Calculation according to the dielectric model.

x-axis caption: "Soil temperature"

**Reference to Fig. 2**

Mironov V.L., Muzalevskiy K.V. Measurement method of the temperature dependence of the dielectric permittivity of topsoil on the Yamal Peninsula using data of the radiometer MIRAS SMOS// Regional problems

of remote sensing of the Earth. 2014. C. 186-189. (in Russian). Available online: http://rprs.sfu-kras.ru/sites/default/files/v_pechat_-materialy_mezhdunarodnoy_nauchnoy_konferencii.pdf (p. 186-189)

[Figure]

**Reference to Fig. 12**
Roy A. et al. Response of L-Band brightness temperatures to freeze/thaw and snow dynamics in a prairie environment from ground-based radiometer measurements, Remote Sensing of Environment, Volume 191, 2017, Pages 67-80.

**Fig. 12.** Comparison of retrieved ground permittivity ($\varepsilon_{G,ret}$) with soil temperature at Scene 1 (blue) and Scene 3 (green). The dotted line shows the optimized threshold value of $\varepsilon_{G,ret}$ (7.65) for F/T retrieval.

[Figure]

**Figure 3.** In situ measurements of soil temperature **(a)** and dielectric constant **(b)** for a single SoilSCAPE sensor node (S6) at the Prudhoe Meadow site in northern Alaska (http://soilscape.usc.edu, last access: 1 May 2018).

**Reference to Fig. 3**

Tao, J., Koster, R. D., Reichle, R. H., Forman, B. A., Xue, Y., Chen, R. H., and Moghaddam, M.: Permafrost variability over the Northern Hemisphere based on the MERRA-2 reanalysis, The Cryosphere, 13, 2087–2110, https://doi.org/10.5194/tc-13-2087-2019, 2019.

These phenomena have been known for a long time ago.

[Figure]

**Refefence to Fig. 80**
Dostovalov B.N., Kudryavtsev V.A. Obshchee merzlotovenie [Permafrost. General physical basics]. Publishing House of Moscow State University, Moscow, 1967. 404p. (In Russian)

[Fig. 80. Measured dielectric constant of soil with different moisture, $W$, at 1 MHz. 1-clay ($W$=35.5%), 2-fine sand (9%), 3- fine sand ($W$=3%)]

However, very few weather stations measure permittivity, and the permittivity data are not publicly available as a final product. As a result, there is currently no technical possibility to validate our algorithm at a representative number of meteorological stations using permittivity data. For this reason, we use soil temperature data to test the algorithm.

3. The book (Ulaby et al, 2014) indicated by Vasiliy Tikhonov (PhD) is actually an encyclopedia in the field of remote sensing. The material contained in this book contains a lot of data from previously published in 1981-1986 in three volumes ("Microwave Remote Sensing: Active and Passive") in collaboration with Moore R.K. and Fung A.K. These books are well known to us and are classics for most specialists in the world. The authors do not understand what particular segment of the text Vasiliy Tikhonov (PhD) had in mind when referring to encyclopedic book in 1013 pages. At the same time, in paragraph 4-8, devoted to the dielectric constant of the soil, we did not find the information pointed out by Vasiliy Tikhonov (PhD). In paragraph 4-8.2 there is no information that the soil can be in a thawed state at negative temperatures and this state depends on the texture of soils.

The dependence of $\varepsilon_{soil}$ on temperature was examined by Hoekstra and Delaney (1974) at 10 GHz and by Hallikainen et al. (1984b) over a wide frequency range extending from 3 GHz to 37 GHz. Above 0 °C, $\varepsilon'_{soil}$ and $\varepsilon''_{soil}$ are weakly dependent on temperature, but as $T$ crosses the freezing temperature of water, both $\varepsilon'_{soil}$ (Ulaby et al, 2014, p.151.)

and $\varepsilon''_{soil}$ change markedly, as illustrated by the data in Fig. 4-35. (Ulaby et al, 2014, p.152.)

Vasiliy Tikhonov (PhD) cites the well-known book (Ulaby et al, 2014), but unfortunately it contains information about soil permittivity and models of soil permittivity relevant for 1980-1995 and does not reflect the current state of the issue.

Numerous studies have been conducted to determine the dielectric behavior of soil-water mixtures, most notably those reported by Wang et al. (1980), Dobson et al. (1985), Hallikainen et al. (1985), Roth et al. (1990), and Peplinski et al. (1995). The measured dielectric data (Ulaby et al, 2014, p.150.)

**Reference**

Ulaby, F. T., Long, D. G., Blackwell, W. J., Elachi, C., Fung, A. K., Ruf, C., Sarabandi, K., Zebker, H. A., and Van Zyl, J. Microwave radar and radiometric remote sensing, University of Michigan Press Ann Arbor, MI, USA, 2014.

Note that the Dobson model contained in the (Ulaby et al, 2014) was created for positive soil temperatures. Later, the Dobson model was modified for negative temperatures (Zhang et al, 2003).

**Reference**

Zhang, L.; Shi, J.; Zhang, Z.; Zhao, K. The estimation of dielectric constant of frozen soil-water mixture at microwave bands. In Proceedings of the 2003 IEEE International Geoscience and Remote Sensing Symposium, Toulouse, France, 21–25 July 2003; Volume 4, pp. 2903–290.

Later, Mironov (of which I am a learner) a model was created (Mironov et al, 2013), which was found to be more accurate and is currently chosen as the main dielectric model in the SMAP and SMOS satellite algorithms (Mialon et al, 2015; Wigneron et al, 2017) to soil moisture retrieval in a global scale. Numerous dielectric models have been developed for organic and mineral soils at sub-zero temperatures by the Mironov's group. The authors of the article (Bircher et al, 2016, p. 15.) especially pointed to the success of the Mironov's group in the field of creating temperature-dependent dielectric models:

one. The temperature-dependent Mironov et al. dielectric models for organic substrates should be exploited for use in satellite data applications where negative temperatures are one of the major drivers (e.g., freeze-thaw, permafrost or snow-related products). Ideally, in the future, common efforts with the research team of V. Mironov should be undertaken to merge the advantages of the simple empirical approach and the Mironov et al. organic models.

**References**

Bircher, S.; Demontoux, F.; Razafindratsima, S.; Zakharova, E.; Drusch, M.; Wigneron, J.-P.; Kerr, Y.H. L-Band Relative Permittivity of Organic Soil Surface Layers—A New Dataset of Resonant Cavity Measurements and Model Evaluation. Remote Sens. 2016, 8, 1024. https://doi.org/10.3390/rs8121024

Mialon A. et al., "Comparison of Dobson and Mironov Dielectric Models in the SMOS Soil Moisture Retrieval Algorithm," in IEEE Transactions on Geoscience and Remote Sensing, vol. 53, no. 6, pp. 3084-3094, June 2015, doi: 10.1109/TGRS.2014.2368585.

Mironov V., Y. Kerr, J. Wigneron, L. Kosolapova and F. Demontoux, "Temperature- and Texture-Dependent Dielectric Model for Moist Soils at 1.4 GHz," in IEEE Geoscience and Remote Sensing Letters, vol. 10, no. 3, pp. 419-423, May 2013, doi: 10.1109/LGRS.2012.2207878.

Wigneron J.-P., T.J. Jackson, P. O'Neill, G. De Lannoy, P. de Rosnay, J.P. Walker, P. Ferrazzoli, V. Mironov, S. Bircher, J.P. Grant, M. Kurum, M. Schwank, J. Munoz-Sabater, N. Das, A. Royer, A. Al-Yaari, A. Al Bitar, R. Fernandez-Moran, H. Lawrence, A. Mialon, M. Parrens, P. Richaume, S. Delwart, Y. Kerr, Modelling the passive microwave signature from land surfaces: A review of recent results and application to the L-band SMOS & SMAP soil moisture retrieval algorithms, Remote Sensing of Environment, Volume 192, 2017, Pages 238-262.

4. What is thawed or frozen soil? The authors believe that this well-known term, especially for the readers of The Cryosphere, does not need to be defined in the text of our manuscript. The authors of the

manuscript understand the frozen soil as a commonly used definition: Soil or rock in which part or all of the *pore water* consists of ice (Harris, 1988, p. 39).

**Reference**

Harris S.A., H.M. French, J.A. Heginbottom, G.H. Johnston, B. Ladanyi, D.C. Sego, R.O. van Everdingen National Research Council of Canada Ottawa, Ontario, Canada KIA OR6. 1988.

The temperature at which ground freezing starts may be lower than 0°C due to *freezing-point depression*. A) We have no information that the soil was saline in our test sites. B) Under laboratory conditions (in measuring cells), we and other authors observed supercooling of water in the soil freezing cycle. In the measuring cells, the conditions of soil water phase transitions can be observed over a wide range of sub-zero temperatures, e.g. at -7°C, -5°C (freezing cycle) (Mironov et al, 2010).

[Figure]

Fig. 5. Behavior of the reduced characteristics for [(a) and (b)] the RI $(n_s - 1)/\rho_d$ and (c) NAC $\kappa_s/\rho_d$ versus temperature at different values of gravimetric moisture in the case of shrub terrain soil, at a frequency of 7.8 GHz. The lines are aids to the eye.

**Reference to Fig. 5**
Mironov V. L., R. D. De Roo and I. V. Savin, "Temperature-Dependable Microwave Dielectric Model for an Arctic Soil," in IEEE Transactions on Geoscience and Remote Sensing, vol. 48, no. 6, pp. 2544-2556, June 2010, doi: 10.1109/TGRS.2010.2040034.

At the same time, in the thawing cycle, the soil becomes thawed when passing through 0C, the hysteresis of dielectric constant can be observed (Lukin et al, 2008, Fig. 1; He et al, 2013, Fig.14).

[Figure]

[Fig.1. Static permittyvity (1 – Free water, 2 – Bound water)]

**Reference to Fig. 1**
Lukin Yu. I., V. L. Mironov, S. A. Komarov. Investigation of the dielectric spectra of wet soil during freezing-thawing // Izvestiya vuzov. Physica. — 2008. — Vol. 51. No. 9. —P. 24–28.

More pronounced hysteresis can also be observed if, during the measurements at each soil temperature, the thermodynamic equilibrium is not sufficiently controlled (Mavrovic et al, 2021, Fig. 5).

[Figure]

Fig. 14. Soil freezing and thawing curves showing hysteresis measured on Mundare loamy sand (described in Table 1) with two different initial water contents: (A, C) measured permittivity, (B, D) estimated liquid water content using the confocal model (parameters in Table 3).

**Reference to Fig. 14**

He H., Dyck M. Application of Multiphase Dielectric Mixing Models for Understanding the Effective Dielectric Permittivity of Frozen Soils. Vadose Zone Journal. 2013. 10.2136/vzj2012.0060

[Figure]

**Figure 5.** Real ($\varepsilon'$) and imaginary ($\varepsilon''$) permittivity of an organic soil sample from the Old Black Spruce site (see Table 1) during freeze–thaw cycles in a cold chamber environment. The OECP and HP instruments monitored soil permittivity, while TD GRMDM and Zhang are model results. The hysteresis effect displayed here is amplified by the experimental setup (discussed in the text). Experiment conducted from 1 to 7 February 2018.

**Reference to Fig. 5**

Mavrovic A., Renato Pardo Lara, Aaron Berg, François Demontoux, Alain Royer, Alexandre Roy. Soil dielectric characterization during freeze–thaw transitions using L-band coaxial and soil moisture probes. Hydrol. Earth Syst. Sci., 25, 1117–1131, 2021. https://doi.org/10.5194/hess-25-1117-2021

However, for dielectric sensors in real field conditions, weather station data provide an estimate for soil freeze/thaw temperatures near 0°C (Lara et al, 2020, Fig. 7) within the error of temperature measurement by modern digital sensors ±0.5°C:

transitions, the permittivity SFC/STC was identified. The freezing and melting point depression for the network was estimated as $T_{f/m} = -0.35 \pm 0.2$, with $T_f = -0.41 \pm 0.22$ °C and $T_m = -0.29 \pm 0.16$ °C, respectively.

[Figure]

**Figure 7.** Sample data from Kenaston Network station 19 during the first freeze-thaw event of 2013. Soil permittivity for freezing (blue) and thawing (red) transition measurements as a function of time (a) and soil temperature (b) from a Cambridge sandy loam sample. The logistic models fitted on the data were used to estimate the freezing/melting point depression, $T_{f/m}$, and associated temperature dispersion, $s_{f/m}$. These values correspond to cumulative probabilities of 0.27, 0.50, and 0.73 from the associated logistic distribution. We interpret these values as an estimate of the relative frozen soil moisture fraction at a given minimal temperature.

**Reference to Fig.7**

Lara, R. P., Berg, A. A., Warland, J., Tetlock, E. (2020). In situ estimates of freezing/melting point depression in agricultural soils using permittivity and temperature measurements. Water Resources Research, 56, e2019WR026020. https://doi.org/10.1029/2019WR026020

For example, if it turn to the weather station located at the Toolik lake, Noth Slope of Alaska, (STATION: AK301, TOOLIK LAKE LTER (301)), which measured the dielectric constant (capacitance sensor Vitel) and temperature of soil (CALM_Data, 2022), the hysteresis of permittivity is also not observed for soils under natural conditions in the area of the weather station. And the freezing and thawing point depression is near 0°C (take into account the error of soil temperature measuring). Below based on the data (see CALM_Data, 2022; columns ER(9cm), EI(9cm), TEMP(deg C) in the files Toolik_00ave.xls, Toolik_99ave.xls, Toolik_98ave.xls) is depicted figure.

[Figure]

Figure shows real part of soil permittivity vs soil temperature at the depth of 9cm measured from 1998-2000 by the Toolik lake station.

**Reference to data**

CALM_Data. Available online, 2022:
https://www2.gwu.edu/~calm/data/CALM_Data/North%20America/Alaska/North%20Slope/U12_toolik/

Apparently, in natural (real), but not in laboratory, conditions, there are too many random factors that affect the increase in the probability of the formation of crystallization zones immediately when the soil temperature falls below -0°C. The phase transition itself is the sharper, than the lower the content of clay fraction (Mironov et al, 2017, Fig. 4) (organic matter or Specific surface area of soil- assumption). See also figure above (He et al, 2013, Fig. 14).

[Figure]

**Fig. 4.** Comparison of CRPs between calculated and measured values for basic soils. The CRPs of moist soil as a function of temperature at the fixed volumetric moistures $m_v$ (given by inscriptions). a) The CRPs of soil 1, b) the CRPs of soil 2, c) the CRPs of soil 3. The measured CRP values are represented by symbols. The solid lines correspond to the CRPs estimated with the developed model ($T < 0$) and earlier developed model ($T \geq 0$). The dash lines correspond to the CRPs calculated with the Zhang model ($T < 0$) and Dobson ($T \geq 0$).

**Reference to Fig. 4**

(Mironov et al, 2017) Mironov et al. Temperature- and texture-dependent dielectric model for frozen and thawed mineral soils at a frequency of 1.4 GHz. REMOTE SENSING OF ENVIRONMENT. 2017. Vol. 200. P. 240-249; 10.1016/j.rse.2017.08.007

At some low moisture levels, for example, for mineral soil of <9% (by volume) (Mironov et al, 2017, Fig. 4, see above) or organic soil 0.338 g/g (by weight) (Mironov et al, 2010, Fig.5, see above), ice does not form in the soil at all. (No heat of crystallization is released in calorimetric measurements, as shown for Na-bentonite clay when compared with the dielectric method (Mironov et al, 2018).)

**Reference**

(Mironov et al, 2018) Mironov V.L., A.Yu Karavayskiy, Yu.I. Lukin, Pogoreltsev E.I. Joint studies of water phase transitions in Na-bentonite clay by calorimetric and dielectric methods. Cold Regions Science and Technology. 2018. Vol. 153. P. 172-180.

5. It is known that Arctic soils are almost always waterlogged, and their moisture content in summer from year to year is approximately the same for specific test site. In (Muskett et al, 2014), this is well demonstrated by the Romanovsky group.

[Figure]

**Figure 4.** (a) Daily time series comparisons of soil moisture contents at sites in Alaska and Russia by ground measurement and NASA- and JAXA-algorithm retrievals; (b) Daily time series comparisons of soil moisture contents at sites in Alaska and Russia by measurement and NASA- and JAXA-algorithm retrievals.

[Figure]

**Figure 4.** (a) Daily time series comparisons of soil moisture contents at sites in Alaska and Russia by ground measurement and NASA- and JAXA-algorithm retrievals; (b) Daily time series comparisons of soil moisture contents at sites in Alaska and Russia by measurement and NASA- and JAXA-algorithm retrievals.

**Reference to Fig. 4**

(Muskett et al, 2014) Muskett, R. , Romanovsky, V. , Cable, W. and Kholodov, A. (2015) Active-Layer Soil Moisture Content Regional Variations in Alaska and Russia by Ground-Based and Satellite-Based Methods, 2002 through 2014. International Journal of Geosciences, 6, 12-41. doi: 10.4236/ijg.2015.61002.

The volumetric soil moisture at the test sites of Alaska (used in our work) in the summer before freezing is about W=35-44%. This moisture content exceeds the maximum amount of bound water in organic soil. (By bound water we here mean tightly bound water and osmotically bound water due to the diffuse part of the double electrical layer - transition bound water.) So for an organic soil with a dry bulk density of

0.256-0.6 g/cm$^3$, the volumetric content of the maximum amount of bound water can reach Wt= 0.1-0.24 cm$^3$/cm$^3$ (estimated based on the maximum amount of bound water by weight of 0.4g/g, see (Mironov et al., 2010).

**Reference**

(Mironov et al, 2010) Mironov V. L., R. D. De Roo and I. V. Savin, "Temperature-Dependable Microwave Dielectric Model for an Arctic Soil," in IEEE Transactions on Geoscience and Remote Sensing, vol. 48, no. 6, pp. 2544-2556, June 2010, doi: 10.1109/TGRS.2010.2040034.

For this reason, for most Arctic test areas, when the temperature drops below -0C, ice is formed in the soil from free soil water in the amount of Wice=W-Wt(Ts=-0C). (Here we do not take into account the migration of water to the freezing front, etc.) At the same time, the dielectric constant decreases abruptly as a result of the first kind phase transition of soil water, and the soil is considered to be frozen.

[Figure]

Figure shows real part of soil permittivity vs soil temperature at the depth of 9cm measured from 1998-2000 by the Toolik lake station.

**Reference**

CALM_Data. Available online, 2022:
https://www2.gwu.edu/~calm/data/CALM_Data/North%20America/Alaska/North%20Slope/U12_toolik/

Otherwise, if total soil moisture W, W< Wt(Ts=-0°C), ice does not form at Ts=-0°C, and there is only unfrozen water in the soil in the amount of W. In this case, the soil is not considered frozen. Ice will begin to form in the soil below such a negative temperature, Ts_depression, at which the equation W=Wt(Ts=Ts_depression) is satisfied. The low moisture content of test sites in Finland before soil freezing (varies on average from about ~15% to 40% (Ikonen, 2016, 2018)) can be the same or less than the value of the maximum amount of bound water estimated above (Wt= 0.1-0.24 cm$^3$/cm$^3$).

**References**

Ikonen, J., Vehviläinen, J., Rautiainen, K., Smolander, T., Lemmetyinen, J., Bircher, S., and Pulliainen, J.: The Sodankylä in situ soil moisture observation network: an example application of ESA CCI soil moisture product evaluation, Geosci. Instrum. Method. Data Syst., 5, 95–108, https://doi.org/10.5194/gi-5-95-2016, 2016.

Ikonen, Jaakko, Tuomo Smolander, Kimmo Rautiainen, Juval Cohen, Juha Lemmetyinen, Miia Salminen, and Jouni Pulliainen. 2018. "Spatially Distributed Evaluation of ESA CCI Soil Moisture Products in a Northern Boreal Forest Environment" Geosciences 8, no. 2: 51. https://doi.org/10.3390/geosciences8020051

As a result, for test sits (SO and SA), the temperature of phase transition processes may be lower than 0°C, or ice may not form in the soil at all at -0°C. (On the other hand, the process of freezing can be slowed down by undergrowth with snow cover, screening the soil.) In our manuscript, we did not remove the data (SO and SA) from the general analysis. Information about the frozen state in these areas was not taken into account in the overall error assessment of the method, as it gave a large error. This is not a drawback of the method, but, on the contrary, indicates the sensitivity of our method to such test sites where the soil is in an unstable-transitional state, which cannot be said to be in a thawed or frozen state. From our point of view, this is an advantage, not a disadvantage of our method.

**Comment 3.**  The authors determine the effective temperature of soil using the AMSR2 6.9 GHz vertical polarization data. This is allegedly based on the assumption that for this frequency, the AMSR2 sounding angle (55 deg) corresponds to soil Brewster angle. The Authors are probably unaware that the Brewster angle of a soil is determined by its moisture and can vary quite widely (see, e.g., Ulaby, Long, 2014). Hence, it is wrong to arbitrarily assume the 55-degree angle to be the soil Brewster angle.

**Response to comment 3.** We do not understand which particular segment of the text Vasiliy Tikhonov (PhD) had in mind referring to the book (Ulaby et al, 2014) of 1013 pages. For a moist soil with rough surface, the Brewster angle is 57° (Ulaby, 2014, see Fig. 10-17 and text on page 437, and addition Fig. 12-5 ), which is very close to the AMSR2 viewing angle of 55°.

**For greater objectivity**, in our answer to Vasiliy Tikhonov (PhD), we note that the book (Ulaby, 2014) contains Fig. 2-17, which shows the results of calculating the modulus of the reflection coefficient from ideally smooth dry e1=3 and wet e1=25 soil, not covered with snow or vegetation.

1.      The real soil surface is always rough surface, and as the roughness increases, the Brewster angle decreases (Ulaby, 2014, Fig. 10-17 and text on p. 437):

> "…Brewster angle [changes] from about 60° for the smoothest surface with s = 0.246 cm (or ks = 0.515) to about 57° for the roughest surface with s = 0.926 cm (or ks = 1.94)."

As a result, the AMSR2 viewing angle of 55° can be very close to the Brewster angle for a real rough surface (57° for the roughest surface with s = 0.926 cm).

[Figure]

**Figure 10-17:** Comparison of I²EM-computed bistatic scattering coefficient with measurements made in the incidence plane ($\theta_i = \theta_s$ and $\phi = 0$) for three surfaces with different roughnesses.

[Figure]

**Figure 12-5:** Measured brightness temperatures as a function of incidence angle at (a) 1.4 GHz, (b) 5 GHz, and (c) 10.7 GHz. The soil temperature for both smooth and rough fields was ~ 20 °C. The volumetric soil moisture content for the smooth field was ~ 0.25 g/cm³ and for the rough field was ~ 0.26 g/cm³ in the top 0 to 10 cm layer [Wang et al., 1983].

**Reference to figures**
Ulaby, F. T., Long, D. G., Blackwell, W. J., Elachi, C., Fung, A. K., Ruf, C., Sarabandi, K., Zebker, H. A., and Van Zyl, J. Microwave radar and radiometric remote sensing, University of Michigan Press Ann Arbor, MI, USA, 2014.

[Figure]

**Figure 2-17:** Plots for $|\rho_h|$ and $|\rho_v|$ as a function of $\theta_1$ for a dry soil surface, a wet-soil surface, and a water surface. For each surface, $|\rho_v| = 0$ at the Brewster angle.

2.        The natural soil is not bare, but covered with snow or vegetation. As a result of interference in the canopy, the region of the Brewster angle is blurred and may contain many minima. This can be seen, for example in (Rodriguez-Alvarez, fig. 7, 8). The Brewster angle for vegetated soil can even be as high as 40-50° (GNSS satellite elevation) or 50-40° (in viewing angle) (Rodriguez-Alvarez, 2011, Figs. 7, 8).

[Figure]

Fig. 7.  Simpler model, vegetation-covered soils case. Simulated interference power received versus reflectivity. (a) Bare soil produces one notch, (b) 60-cm vegetation layer + soil layer produces three notches, and (c) 90-cm vegetation layer + soil layer produces four notches. Note that first notch is due to the Brewster's angle, but new notches appear due to oscillations in the reflectivity.

**Reference to Fig.7**

Rodriguez-Alvarez N. *et al*., "Land Geophysical Parameters Retrieval Using the Interference Pattern GNSS-R Technique," in *IEEE Transactions on Geoscience and Remote Sensing*, vol. 49, no. 1, pp. 71-84, Jan. 2011, doi: 10.1109/TGRS.2010.2049023.

Also, experimentally measured values of brightness temperature over thawed or frozen tundra soil show that the maximum brightness temperature on vertical polarization is observed in the range of angles 35-70° (taking into account the measurement error of satellite radiometers) (Lemmetyinen et al, 2016, Fig.5-7;  Ulaby et al, 2014, Fig. 12-40).

Brewster angle is 55-65°                          Brewster angle is 45-55°

[Figure]

**Fig. 5.** Simulated (lines) and measured (symbols) L-band brightness temperatures $T_B^p(\theta_k)$ ($p$ = H, V) as function of the incidence angle $\theta_k$ (solid lines: horizontal polarization ($p$ = H); dashed lines: vertical polarization ($p$ = V). Simulations using the indicated snow densities $\rho_S$ are shown in comparison with tower-based ELBARA-II observations (triangles) performed at the forest clearing site. All simulations were made for the indicated ground permittivity $\varepsilon_G$ and ground temperature $T_G$ measured from SO11 sensors at 5 cm depth.

**Fig. 6.** Simulated (lines and shaded areas) and measured (triangles) $T_B^p(\theta_k)$ for the snow-covered footprints at the wetland site and for the corresponding footprints after snow clearing. Extent of snow clearance indicated with vertical dashed lines.

Brewster angle are a) 60-70°, b) 55-70°, c) 55-65°, d) 35-65°

[Figure]

**Fig. 7.** Examples of simulated (lines) and observed (triangles) brightness temperatures $T_B^P(\theta_k)$ using the indicated retrieved parameters $\mathbf{P} = (\varepsilon_{G,ret}, \rho_{S,ret})$. Forest clearing site for summer (a) and winter (b); wetland site for late autumn (c) and winter (d). Observation angle ranges of $40° \leq \theta_k \leq 60°$ and $50° \leq \theta_k \leq 60°$ used for retrievals over the forest clearing and wetland sites, respectively. Hollow triangles indicate snow free conditions, filled triangles indicate presence of snow.

**Reference to Fig. 7**

Lemmetyinen J., Mike Schwank, Kimmo Rautiainen, Anna Kontu, Tiina Parkkinen, Christian Mätzler, Andreas Wiesmann, Urs Wegmüller, Chris Derksen, Peter Toose, Alexandre Roy, Jouni Pulliainen, Snow density and ground permittivity retrieved from L-band radiometry: Application to experimental data, Remote Sensing of Environment, Volume 180, 2016, Pages 377-391.

Brewster angle is <40-60°

[Figure]

Figure 12-40: Comparison of theory with angular measurements for a snow layer with $\varepsilon_{ice} = 3.15 - j14 \times 10^{-3}$, snow density $\rho_s = 0.3$ g/cm$^3$, equivalent scatterer radius $r = 0.85$ mm, and wetness by weight $m_w = 0.06$ percent [Fung and Eom, 1981].

**Reference to Fig. 12-40**
Ulaby, F. T., Long, D. G., Blackwell, W. J., Elachi, C., Fung, A. K., Ruf, C., Sarabandi, K., Zebker, H. A., and Van Zyl, J. Microwave radar and radiometric remote sensing, University of Michigan Press Ann Arbor, MI, USA, 2014

Our calculations, which we additionally carried out for a homogeneous soil, covered with a layer of snow or vegetation, also confirm that the Brewster angle is blurred and is in the region of the AMSR2 viewing angle (see Fig. 1A, below). In the calculations, the content of organic matter in the soil was set equal to 50% (by weight), soil moisture 45% (by volume). The temperature of thawed and frozen soil was set equal to 20°C and -10°C, respectively. The soil permittivity was calculated at a frequency of 1.4 GHz based on formulas from (Mironov et al, 2019).

**Reference**

Mironov V. L., L. G. Kosolapova, S. V. Fomin and I. V. Savin, "Experimental Analysis and Empirical Model of the Complex Permittivity of Five Organic Soils at 1.4 GHz in the Temperature Range From −30 °C to 25 °C," in IEEE Transactions on Geoscience and Remote Sensing, vol. 57, no. 6, pp. 3778-3787, June 2019, doi: 10.1109/TGRS.2018.2887117.

[Figure]

Figure 1A. Modulus of reflection coefficient for soil covered with vegetation (red) and snow(blue) layer.

The density of dry snow was set equal to 0.32g/cm³, and the effective permittivity of the vegetation cover was taken to be 1.47+0.36i (Schwank, 2014).

**Reference**

Schwank M. et al. Model for microwave emission of a snow-covered ground with focus on L band, Remote Sensing of Environment, Volume 154, 2014, Pages 180-191

3.      We also note the following fact. The emissivity on vertical polarization in the range of angles slightly less than the Brewster angle, weakly depends on the properties of the layer covering the dielectric half-space (with an increase in the thickness of ice over water over 0.6 cm.) (Sirounian, 1968)

[Figure]

Fig. 6. Polarized emissivity of ice on top of water for λ = 10 cm. Polarization maximum decreases and shifts toward small nadir angles with the increase of ice thickness. Also, the polarized emissivity increases with ice thickness at small nadir angles.

**Reference to Fig. 6**
Sirounian, V. The effect of the temperature, angle of observation, salinity, and thin ice on the microwave emission of water. J. Geophys. Res., 1968, Vol. 73, No. 14, p. 4481-4486.

In the case of a frozen soil, the emissivity on vertical polarization in the range of angles slightly less than the Brewster angle is practically independent on the presence or absence of snow on a smooth soil surface, as well as snow of various densities of 150 kg/m$^3$ (snow height 100cm)-300kg/m$^3$ (snow height 50cm) (Schwank et al, 2014).

[Figure]

**Reference to Fig. 9**

Schwank M. et al. Model for microwave emission of a snow-covered ground with focus on L band, Remote Sensing of Environment, Volume 154, 2014, Pages 180-191.
(Also see Fig. 10 in (Schwank et al, 2014).)

**Fig. 9.** Angular dependences of emissivities $E^p(\theta_0)$ of GS systems with frost-depth $d_{FG} =$ 40 cm. Definitions of the SPs are provided in Table 2.

It can also be seen that as the snow density increases, the Brewster angle decreases and tends from 65° to ~57–50°, at which the vertical polarization emissivity does not depend on the properties of the layer covering the soil. (At the same time, the differences in the emissivity of the soil covered with snow of various density and height at angles of 40-55 are very small (see Fig. 9 in (Schwank et al, 2014)). It can be assumed that the above results and similar conclusions are also valid when soil covered with vegetation layer.

As a result, for a real rough soil covered with vegetation or snow, the Brewster angle is very diffuse and its location can be approximately assumed to be near the AMSR2 viewing angle of 55°. In this case, at angles slightly smaller than the Brewster angle, the emissivity on vertical polarization weakly depends on the properties of the layer covering the homogeneous dielectric half-space. In addition, it should be taken into account that soils of different moisture content, height and biomass of vegetation, different height and density of snow fall into the ~50x50 km pixel, which leads to even more blurring and smoothing of the Brewster angle area.

**Comment 4.** Further, literally the next but one sentence reads: "Further, estimates of ΓH(f) will be considered as the apparent values of reflectivity, since the absolute value of TbV(6.9) does not coincide with the actual values of the soil surface temperature Ts0, but is only proportional to them." Well, is it equal or proportional?! And what kind of physical characteristic is "apparent value of reflectivity"? To whom, how and why is it apparent?!

**Response to comment 4.** Our text says that «…surface temperature Ts0 can be estimated based on the measurements of brightness temperature TbV(6.9)…» And as it was proved in the answer to question 3,

there are physical basics for this statement. We do not have a statement with the term "equal" in the text, concerning $T_{s0}$ and TbV(6.9). At the beginning of the logical construction there is a statement that: «The soil surface temperature Ts0 *can be estimated* based on the measurements of brightness temperature TbV(6.9) » (line 120). Thus, we are initially talking not about the exact value, but about the estimate of Ts0. What is it that we have no right to say here? And in manuscript we declared, using the terms of "proportional", "estimated", "considered as the apparent values". We see no error in our statements.

The term "apparent" is in general use and accompanies many terms. Apparent value is the value, which may be different from actual one (e.g. apparent impedance). Dear, Vasiliy Tikhonov (PhD), see (The Authoritative Dictionary of IEEE Standards Terms, 2000).

**Reference**

"The Authoritative Dictionary of IEEE Standards Terms, Seventh Edition," in IEEE Std 100-2000 , vol., no., pp.1-1362, 11 Dec. 2000, doi: 10.1109/IEEESTD.2000.322230.).

Dear, Vasiliy Tikhonov (PhD) also see caption to the Fig. 4.13 in book (Bogorodsky et al, 1985)

[Figure]

Figure caption is "Fig. 4.13. Dependence of the ***apparent*** degree of polarization on the angle of observation." (in Russian).

**Reference**

Bogorodsky V.V., Kozlov A.I. Mikrovolnovaya radiometriya zemnyh pokrovov. [Microwave radiometry of earth covers]. Gidrometeoizdat. Leningrad. 1985. 272p.

**Comment 5.** It is absolutely unclear why the Authors use microwave radiometry data to determine effective soil temperature. There are much more effective methods that use infrared data. At present, methods using satellite microwave radiometry to determine soil temperature are still under development and are not finalized yet. I recommend the Authors the review of Duan et al. (2020) on this topic.

**Response to comment 5.** We are familiar with the article (Duan et al., 2020). We cannot find information in the article (Duan et al., 2020) that would indicate the possibility of measuring soil temperature using infrared (IR) radiometers under canopy. We did not use thermal infrared images. First, IR sensors are measured Land surface temperature (LST) but not soil. IR sensors measures skin temperatures at the surface materials-atmosphere interface (Hachem, 2012), but not soil.

> **Reference**
>
> Hachem, S., Duguay, C. R., and Allard, M.: Comparison of MODIS-derived land surface temperatures with ground surface and air temperature measurements in continuous permafrost terrain, The Cryosphere, 6, 51–69, https://doi.org/10.5194/tc-6-51-2012, 2012.

Second, IR sensors are of limited use in polar regions because these areas are dark (polar night) for half the year and are often cloud covered (Ulaby et al, 2014, p.913).

> **Reference**
>
> Ulaby, F. T., Long, D. G., Blackwell, W. J., Elachi, C., Fung, A. K., Ruf, C., Sarabandi, K., Zebker, H. A., and Van Zyl, J. Microwave radar and radiometric remote sensing, University of Michigan Press Ann Arbor, MI, USA, 2014

Third, LSTs (temperatures at the surface materials-atmosphere interface) are found to be better correlated with Tair (1–3 m above the ground) than with soil temperature (3–5 cm below the ground surface) (Hachem, 2012, Muzalevskiy 2016).

[Figure]

**Fig. 5.** Comparison between the mean daily LST (combined Day/Night/Terra/Aqua) and mean daily air temperature at West Dock (WD1) for **(a)** complete year, **(b)** winter, and **(c)** summer. On left, LST overlayed on meteorological station measurements, and on right relation between the two sets of measurements.

[Figure]

**Fig. 2.** Relation between mean daily MODIS LST (combined Terra and Aqua) and mean daily GST at **(a)** Betty Pingo Upland (BPU1), **(b)** Franklin Bluff (FB1), **(c)** Western Kuparuk (WK1), and **(d)** Sila (Sila1).

**Reference to Fig. 5 and Fig. 2**

(Hachem et al, 2012) Hachem, S., Duguay, C. R., and Allard, M.: Comparison of MODIS-derived land surface temperatures with ground surface and air temperature measurements in continuous permafrost terrain, The Cryosphere, 6, 51–69, https://doi.org/10.5194/tc-6-51-2012, 2012.

[Figure]

Fig. 2. Correlation between average daily air temperature and 8 A.M. air temperature. RMSE and $R^2$ values were determined to be 2.6 °C and 0.96, respectively.

[Figure]

[Figure]

Fig. 5. Comparison between average daily soil temperatures measured at five *in situ* sites (WS-AST) and average MODIS LST values on the test site by combining day and night MODIS products (V041 MODIS LST L3 Global 1km) during 2013 year. (a) Time series and (b) correlation between these values. The calculated RMSE are 4.72 and 7.58 °C, respectively, for MODIS LST versus air temperature and MODIS LST versus WS-AST. The calculated determination coefficient are 0.88 and 0.38, respectively, for these cases.

**Reference to Fig. 2 and 5**

(Muzalevskiy et al, 2016) Muzalevskiy K. V. and Z. Ruzicka, "Retrieving Soil Temperature at a Test Site on the Yamal Peninsula Based on the SMOS Brightness Temperature Observations," in *IEEE Journal of Selected Topics in Applied Earth Observations and Remote Sensing*, vol. 9, no. 6, pp. 2468-2477, June 2016, doi: 10.1109/JSTARS.2016.2553220.

**Comment 6.** The Authors analyze brightness temperature of the test sites obtained by two satellite sensors at different viewing angles: SMAP at 40 degrees and GCOM-W1 at 55 degrees. Thus, there is a comparison of brightness temperatures of different bands received at different angles: band 1.4 GHz at 40 degrees, and bands 6.9 GHz, 10.7 GHz, 18.7 GHz, 36.5 GHz at 55 degrees. And then the conclusion is made about the efficiency of bands 1.4 and 6.9 GHz, and bands 1.4 and 36.5 GHz. The question arises, "Have the authors heard anything about Fresnel formulas?" At different angles, the reflectivity of the same surface is different. The Authors operate with incomparable characteristics.

**Response to comment 6.** Dear, Vasiliy Tikhonov (PhD), unfortunately, the solution of the Maxwell equation for layered inhomogeneous dielectric half-spaces, with a rough soil boundary covered with snow and vegetation, does not lead to reflection coefficients in the form of Fresnel formulas. The authors of the manuscript have already partially answered on this question to Vasiliy Tikhonov (PhD) above. Vasiliy

Tikhonov (PhD) incorrectly attempts to compare the processes of radiation from an absolutely smooth surface and the radiation of a rough soil surface covered with vegetation or snow. In particular, for example, for a flat-layered dielectric half-space with an Epstein transition layer, the reflection coefficient at horizontal polarization is expressed using gamma-functions (Brekhovskikh, 1960):

$$V = \frac{\Gamma(iS\cos\vartheta_0)\,\Gamma\{-i(S/2)\,[\cos\vartheta_0 + \sqrt{(\cos^2\vartheta_0 - N)}]\}}{\Gamma(-iS\cos\vartheta_0)\,\Gamma\{i(S/2)\,[\cos\vartheta_0 - \sqrt{(\cos^2\vartheta_0 - N)}]\}}$$
$$\times \frac{\Gamma\{1 - i(S/2)\,[\cos\vartheta_0 + \sqrt{(\cos^2\vartheta_0 - N)}]\}}{\Gamma\{1 + i(S/2)\,[\cos\vartheta_0 - \sqrt{(\cos^2\vartheta_0 - N)}]\}}. \quad (14.41)$$

and in the case of a linear permittivity profile, the reflection coefficient is expressed in terms of the Bessel or Hankel functions of the first kind of the fractional index:

$$V = \frac{J_{-\frac{2}{3}} - J_{\frac{2}{3}} + i(J_{\frac{1}{3}} + J_{-\frac{1}{3}})}{J_{-\frac{2}{3}} - J_{\frac{2}{3}} - i(J_{\frac{1}{3}} + J_{-\frac{1}{3}})}. \quad (15.15)$$

$$V = \frac{iH_{\frac{1}{3}}^{(1)}(w_0) - H_{-\frac{1}{3}}^{(1)}(w_0)}{iH_{\frac{1}{3}}^{(1)}(w_0) + H_{-\frac{1}{3}}^{(1)}(w_0)}. \quad (15.19)$$

Reference

Brekhovskikh L.M. Waves in Layered Media, NewYork, NY, USA: Academic. 1960. P. 561.

And for example (Bogorodsky et al, 1977), for the exponential layer

$$\varepsilon_3 = Ne^{-2az} \quad (4.2)$$

$$N = \varepsilon_2 \left(\frac{\varepsilon_2'}{\varepsilon_4}\right)^{d_2/d_3}; \quad a = \frac{1}{2d_3}\ln\left(\frac{\varepsilon_4'}{\varepsilon_2}\right). \quad (4.3)$$

the reflection coefficient on the horizontal polarization has the form

$$R_{\text{rn}} = \frac{\begin{aligned}&\left[\frac{dJ_s(x)}{dz}\frac{1}{ik\cos\beta} - J_s(x)\right]\times\\&\times\left[\frac{dN_s(x)}{dz}\frac{1}{ik\sqrt{\varepsilon_{III} - \sin^2\beta}} - N_s(y)\right] -\\&-\left[\frac{dJ_s(y)}{dz}\frac{1}{ik\sqrt{\varepsilon_{III} - \sin^2\beta}} - J_s(y)\right]\times\\&\times\left[\frac{dN_s(x)}{dz}\frac{1}{ik\cos\beta} - N_s(x)\right]\end{aligned}}{\begin{aligned}&\left[\frac{dJ_s(x)}{dz}\frac{1}{ik\cos\beta} + J_s(x)\right]\times\\&\times\left[\frac{dN_s(x)}{dz}\frac{1}{ik\sqrt{\varepsilon_{III} - \sin^2\beta}} - N_s(y)\right] -\\&-\left[\frac{dJ_s(y)}{dz}\frac{1}{ik\sqrt{\varepsilon_{III} - \sin^2\beta}} - J_s(y)\right]\times\\&\times\left[\frac{dN_s(x)}{dz}\frac{1}{ik\cos\beta} + N_s(x)\right]\end{aligned}}. \quad (4.24)$$

105

on the vertical polarization has the form

$$R_{\text{вп}} = \frac{[Q_1 - J_p(x)]\,[Q_4 - N_p(y)\,e^{ah}] - [Q_3 - J_p(y)\,e^{ah}]\,[Q_2 - N_p(x)]}{[Q_1 + J_p(x)]\,[Q_4 - N_p(y)\,e^{ah}] - [Q_3 - J_p(y)\,e^{ah}]\,[Q_2 + N_p(x)]}, \tag{4.25}$$

$$
\begin{aligned}
Q_1 &= \frac{1}{ik\varepsilon_{\text{II}}\cos\beta}\,\frac{d}{dz}\left[e^{-az}J_p\left(\frac{k\sqrt{N}}{a}\,e^{-az}\right)\right]\Big|_{z=0};\\[6pt]
Q_2 &= \frac{1}{ik\varepsilon_{\text{II}}\cos\beta}\,\frac{d}{dz}\left[e^{-az}N_p\left(\frac{k\sqrt{N}}{a}\,e^{-az}\right)\right]\Big|_{z=0};\\[6pt]
Q_3 &= \frac{1}{ik\sqrt{\varepsilon_{\text{III}} - \sin^2\beta}}\,\frac{d}{dz}\left[e^{-az}J_p\left(\frac{k\sqrt{N}}{a}\,e^{-az}\right)\right]\Big|_{z=-h};\\[6pt]
Q_4 &= \frac{1}{ik\sqrt{\varepsilon_{\text{III}} - \sin^2\beta}}\,\frac{d}{dz}\left[e^{-az}N_p\left(\frac{k\sqrt{N}}{a}\,e^{-az}\right)\right]\Big|_{z=-h};\\[6pt]
p &= \frac{1}{a}\sqrt{a^2 + k^2\sin^2\beta}.
\end{aligned}
\tag{4.26}
$$

$$x = \frac{4\pi\sqrt{\varepsilon_{\text{II}}}}{\ln\dfrac{\varepsilon_{\text{III}}}{\varepsilon_{\text{II}}(0)}}\,\frac{h}{\lambda}\quad \text{при}\ \ z = 0, \tag{4.22}$$

$$y = \frac{4\pi\sqrt{\varepsilon_{\text{III}}}}{\ln\dfrac{\varepsilon_{\text{III}}}{\varepsilon_{\text{II}}(-h)}}\,\frac{h}{\lambda}\quad \text{при}\ \ z = -h. \tag{4.23}$$

**Reference**

Bogorodsky V.V., Kozlov A.I., Tuchkov L.T. Radioteplovoe izluchenie zemnyh pokrovov. [Radiothermal radiation of the earth's covers.] Gidrometeoizdat. Leningrad. 1977. 224p.

Vasiliy Tikhonov (PhD) from the above formulas can make sure that these formulas have nothing to do with the Fresnel formulas. The situation with sensing a real soil with a rough boundary, covered with snow or vegetation, with a compound dielectric profile, apparently, has much more complex patterns than the Fresnel formulas describe.

1.      First, with an increase in the roughness of the soil surface, the angular dependence of the brightness temperature becomes less pronounced, and in the limit of high roughness, the brightness temperature is achieved complete depolarization. On Fig. 12-5 (Ulaby et al, 2014) for rough soil, it can be seen that the variations of brightness temperatures on vertical and horizontal polarizations (between 40-55°) do not much exceed the error, which corresponds to accuracy of measuring, for example, of soil moisture. Let's show it. Indeed, at present, for the algorithms of the SMAP and SMOS satellites, the error in measuring of soil moisture is 4% (Wigneron et al, 2017).

**Reference**

Wigneron J.-P., T.J. Jackson, P. O'Neill, G. De Lannoy, P. de Rosnay, J.P. Walker, P. Ferrazzoli, V. Mironov, S. Bircher, J.P. Grant, M. Kurum, M. Schwank, J. Munoz-Sabater, N. Das, A. Royer, A. Al-Yaari, A. Al Bitar, R. Fernandez-Moran, H. Lawrence, A. Mialon, M. Parrens, P. Richaume, S. Delwart, Y. Kerr, Modelling the passive microwave signature from land surfaces: A review of recent results and application to the L-band SMOS & SMAP soil moisture retrieval algorithms, Remote Sensing of Environment, Volume 192, 2017, Pages 238-262.

[Figure]

**Figure 12-5:** Measured brightness temperatures as a function of incidence angle at (a) 1.4 GHz, (b) 5 GHz, and (c) 10.7 GHz. The soil temperature for both smooth and rough fields was ∼ 20 °C. The volumetric soil moisture content for the smooth field was ∼ 0.25 g/cm³ and for the rough field was ∼ 0.26 g/cm³ in the top 0 to 10 cm layer [Wang et al., 1983].

**Reference to Fig. 12-5**

Ulaby, F. T., Long, D. G., Blackwell, W. J., Elachi, C., Fung, A. K., Ruf, C., Sarabandi, K., Zebker, H. A., and Van Zyl, J. Microwave radar and radiometric remote sensing, University of Michigan Press Ann Arbor, MI, USA, 2014

Experiments and calculations show that for 1% change in soil moisture (let the dry bulk density be equal to 1g/cm3), the brightness temperature changes on average by ~3K (Schmugge,1972; RAO et al, 1987).

[Figure]

Figure 2–Plot of the microwave brightness temperature versus the soil moisture for the 21-cm wavelength radiometer, flight 1 over Phoenix, Ariz., February 25, 1971. X indicates the bare fields; o, the vegetated fields.

**Reference to Fig. 2**

Schmugge T.J. Soil moisture measurements with microwave radiometers. Report. NASA. No. x-652-72-305. 1972

[Figure]

Figure 7. $T_B$ versus EQSM (in weight per cent) for constant soil moisture profile at different frequencies ($T = 300$ K).

**Reference to Fig. 7**

RAO K. S., GIRISH CHANDRA & P. V. NARASIMHA RAO (1987) The relationship between brightness temperature and soil moisture Selection of frequency range for microwave remote sensing, International Journal of Remote Sensing, 8:10, 1531-1545, DOI: 10.1080/01431168708954795

As a result, having an average maximum achievable accuracy of soil moisture measurements using SMAP and SMOS satellites of 4%, we can quite neglect variations in brightness temperatures within 12K! It should also be noted that the emission models (Wigneron, 2011), whose parameters were found on the basis of experimental data, describe the brightness temperatures, measured over bare soil in this experiment, with an error no better than 4–5 K. In the reality of remote sensing from satellite over footprint in tens of km, this error should increase significantly from 4-5K.

> **Reference**
>
> Wigneron J. -P. *et al.*, "Evaluating an Improved Parameterization of the Soil Emission in L-MEB," in *IEEE Transactions on Geoscience and Remote Sensing*, vol. 49, no. 4, pp. 1177-1189, April 2011, doi: 10.1109/TGRS.2010.2075935.

Therefore, when analyzing the above radiometric data, we may well neglect the variations in brightness temperatures in the ranges of 4-12K.

In accordance with expression (3) from the manuscript, Vasiliy Tikhonov (PhD) sees the main problems when calculating

$$\Gamma_p(\theta=40°)=1-Tb_H(\theta=40°)/ Tb_V(\theta=55°),$$

where the observation angles do not coincide when measuring the brightness temperature $Tb_H(\theta=40°)$ and $Tb_V(\theta=55°)$ of SMAP and AMSR2/GCOM-1, respectively. As can be seen from the previous analysis, we have the right to neglect the brightness temperature variations of 4-12K.

Let us turn to the experimentally measured angular values of brightness temperature on horizontal and vertical polarizations over frozen and thawed tundra soil (covered and not covered with snow) (Lemmetyinen et al, 2016, Fig. 7) and soil in Canadian Prairie (Roy et al, 2018, Fig. 3). It can be seen from the figures below that, within the error of 4-12K, the angular dependence of the brightness temperature on the vertical polarization can be neglected within the variation of angles of 40-55°. (The exception is flooded soil.) We believe that at a frequency of 6.9 GHz the trend will be similar, see for example also Fig. 12-5 depicted above from the book (Ulaby et al, 2014).

[Figure]

**Fig. 7.** Examples of simulated (lines) and observed (triangles) brightness temperatures $T_B^P(\theta_k)$ using the indicated retrieved parameters $\mathbf{P} = (\varepsilon_{G,ret}, \rho_{S,ret})$. Forest clearing site for summer (a) and winter (b); wetland site for late autumn (c) and winter (d). Observation angle ranges of $40° \leq \theta_k \leq 60°$ and $50° \leq \theta_k \leq 60°$ used for retrievals over the forest clearing and wetland sites, respectively. Hollow triangles indicate snow free conditions, filled triangles indicate presence of snow.

**Reference to Fig. 7**

Lemmetyinen J., Mike Schwank, Kimmo Rautiainen, Anna Kontu, Tiina Parkkinen, Christian Mätzler, Andreas Wiesmann, Urs Wegmüller, Chris Derksen, Peter Toose, Alexandre Roy, Jouni Pulliainen, Snow density and ground permittivity retrieved from L-band radiometry: Application to experimental data, Remote Sensing of Environment, Volume 180, 2016, Pages 377-391.

[Figure]

**Figure 3.** $T_B$V-pol (blue) and $T_B$H-pol (black) on 4 March 2015 at Scene 1 measured (symbols) and simulated (lines) with the Wave Approach for LOw-frequency MIcrowave emission in Snow (WALOMIS) with added Gaussian noise ($\sigma_d$) to the measured density of 30 kg m$^{-3}$ (**left**), 60 kg m$^{-3}$ (**center**) and 90 kg m$^{-3}$ (**right**).

**Reference to Fig. 3**

Roy A.; Leduc-Leballeur, M.; Picard, G.; Royer, A.; Toose, P.; Derksen, C.; Lemmetyinen, J.; Berg, A.; Rowlandson, T.; Schwank, M. Modelling the L-Band Snow-Covered Surface Emission in a Winter Canadian Prairie Environment. *Remote Sens.* **2018**, *10*, 1451. https://doi.org/10.3390/rs10091451

2.     Secondly, in our work, we propose a semi-empirical approach to identifying the thawed and frozen state of soils. And we don't have requirements to follow mathematical rigor in formulas. We use the formulas as the main highways of radiation laws, in which, due to the empirical approach, we use brightness temperatures that do not exactly match in the sensing angle. In any case, the neglect in the sensing angle that we allowed is contained in the total error of the proposed method.

3.     Thirdly, regardless of the sensing angle, the spatial resolution of SMAP 39x47km (43km on average) and the L1R product of AMSR2/GCOM-W1 with spatial resolution of (35x62km, 48km on average, normalized to the 6.9GHz channel) are very close.

**Comment 7.** On pages 6-7, Authors derive expression (3) for "isothermal and dielectric-homogeneous half-space." However, on page 10, when discussing Figure 3, namely the spectral gradients of brightness temperature and reflectivity, the Authors explain their highest and lowest values by "a significant contrast of temperatures and permittivities between the shallow and deeper emitting layers of soil". Again, one contradicts the other!

**Response to comment 7.** On pages 6-7, our manuscript does not contain the derivation of formulas, and on page 10 there is a page with references. On Fig. 3, the manuscript does not show the gradients of brightness temperature. Equation (3) is not derived in our manuscript. This formula is taken from a published journal article (Muzalevskiy and Ruzicka, 2020) (referenced in the manuscript). The manuscript says that formula (3) coincides with formula (1) if we put in formula (1) $\left.\frac{\Delta T}{\Delta z}\right|_{z=0} = 0$. Formula

(1) is taken from reference (Zuerndorfer and England, 1992). Despite the fact that Vasiliy Tikhonov (PhD) cited incorrect, we consider it necessary to answer, in fact, to the comments of Vasiliy Tikhonov (PhD).

Yes, there seems to be an inaccuracy here that should be corrected based on the text of the article (Muzalevskiy et al., 2021). An earlier version of the article (Muzalevskiy and Ruzicka, 2020) also contains what appears to be a similar inaccuracy, which comes from a search through time for our explanation of the phenomena under study.

For a weakly scattering layer (dry snow, vegetation cover) covering the soil, the brightness temperature Tb can be written with a tau-omega model (Ulaby 12-5, 12-6; Parrens et al, 2017):

$$Tb = (1-\omega)(1-e^{-\tau/\cos\theta})T_{vs} + (1-\omega)(1-e^{-\tau/\cos\theta})\,e^{-\tau/\cos\theta}\,\Gamma(\theta)T_{vs} + (1-\Gamma(\theta))e^{-\tau/\cos\theta}T_{eff}, \qquad (1B)$$

where $\Gamma(\theta) = |R(\theta)|^2 e^{-Hr(\theta)}$ is the soil reflectivity, $R(\theta)$ is the reflection coefficient from layered bare soil with flat surface, $\theta$ is the viewing angle, $Hr(\theta)$ is the soil roughness parameter, $T_{vs}$ is the average temperature of vegetation (snow) cover, $T_{eff}$ is the soil effective temperature, $\tau$ is the optical depth of snow or vegetation cover. Since the temperature of the vegetation (snow) cover and the temperature of the soil are different, there is a temperature gradient. Further, let $T_0$ be the effective temperature of some emitting layer, including lower layers of vegetation or snow and the shallow layers of soil. Then let the approximate equalities be valid $T_{vs} = T_0$, $T_{eff} = T_0$. These equalities do not define an isothermal half-space, but define a half-space in which there is a temperature gradient, and this half-space has an effective temperature $T_0(\theta, f)$. Neglecting the scattering of waves by elements of the vegetation cover and ice crystals in the snow cover $\omega = 0$, (1B) becomes

$$Tb(\theta,f) = (1-|R(\theta,f)|^2 e^{-Hr(\theta,\,f)-\tau(f)/\cos\theta})T_0(\theta,\,f) \qquad (2B)$$

and introducing a new variable $H(\theta,f) = Hr(\theta,f) + \tau(f)/\cos\theta$.

$$Tb(\theta,f) = (1-\Gamma(\theta,f))T_0(\theta,\,f), \quad \text{где } \Gamma(\theta,f) = |R(\theta,f)|^2 e^{-H(\theta,f)}. \qquad (3B)$$

As a result, formula (3B) describes the brightness temperature of a vegetation (snow)-soil layer with an effective temperature $T_0(\theta, f)$, with the combined effect of roughness and optical thickness (vegetation, snow cover) in term of $H(\theta,f)$. We do not find any contradictions in such logic and arguments. And the comments to the formula (3) in manuscript should be correct, in accordance with the above understanding. From the responses to the comments below, it will be clear to what extent $T_0(\theta, f)$ can be related to soil temperature.

**References**

Muzalevskiy, K., Ruzicka, Z., Roy, A., Loranty, M., Vasiliev, A.: Classification of the frozen/thawed surface state of Northern land areas based on SMAP and GCOM-W1 brightness temperature observations at 1.4 GHz and 6.9 GHz, Remote Sensing Letters, 11, 12, 1073-1081, doi: 10.1080/2150704X.2021.1963497, 2021.

Muzalevskiy, K., Ruzicka, Z.: Detection of Soil Freeze/Thaw States in the Arctic Region Based on Combined SMAP and AMSR-2 Radio Brightness Observations, International Journal of Remote Sensing, 41, 14, 5046-5061, doi: 225 10.1080/01431161.2020.1724348, 2020

Parrens M., Jean-Pierre Wigneron, Philippe Richaume, Ahmad Al Bitar, Arnaud Mialon, Roberto Fernandez-Moran, Amen Al-Yaari, Peggy O'Neill, Yann Kerr, Considering combined or separated roughness and vegetation effects in soil moisture retrievals, International Journal of Applied Earth Observation and Geoinformation, Volume 55, 2017, Pages 73-86.

Ulaby, F. T., Long, D. G., Blackwell, W. J., Elachi, C., Fung, A. K., Ruf, C., Sarabandi, K., Zebker, H. A., and Van Zyl, J. Microwave radar and radiometric remote sensing, University of Michigan Press Ann Arbor, MI, USA, 2014

Zuerndorfer, B., England, A. W., Radiobrightness decision criteria for freeze/thaw boundaries, IEEE Transactions on Geoscience and Remote Sensing, 30, 1, 89-102, doi:10.1109/36.124219, 1992.

**Comment 8**. Further the authors engage in formula-juggling, deriving one expression from another. For example, from brightness temperature (with author simplifications) they get surface reflectivity; or from the gradient of brightness temperature spectral density - the gradient of reflectivity. The result is presented in trivial "flip-flop" graphs because one formula follows from the other. In the end, Authors conclude: "Both criteria give comparable accuracies of forecasting thawed and frozen topsoil state for tundra soil cover," which is bluntly obvious, since one formula is derived from the other.

**Response to comment 8.** In contrast to the article (Zuerndorfer and England, 1992), in which it was proposed to use spectral gradients of brightness temperature, in our work it is proposed to use spectral gradients of reflectivity. In addition, our manuscript investigates a wider spectrum of frequencies compared to existing researches, including L-band and C-band.

1. Let there be a value of the brightness temperature measured by a satellite in horizontal polarization at some frequency $Tb_H(f)$. How to determine the emissivity of the underlying surface, if the height and biomass of vegetation (height, snow density), moisture and roughness of the soil surface, as well as the temperature of snow, vegetation and soil, and their temperature profiles are not known? If $Tb_H(f)$ is available and no other information, you will not be able to estimate the reflectivity from the $Tb_H(f)$ value. In addition, an emission model is needed to evaluate reflectivity.

2. In our manuscript, to assess the reflectivity model (3) was used.

$$Tb_p(f)=(1-\Gamma_p(f))T_{s0} \tag{1C}$$

which should be interpreted as the result of the derivation (1B)-(3B), where $T_{s0}$ is the effective temperature of some emitting layer.

3. A simple model (1C) makes it possible to estimate the reflectivity of the emitting layer if its effective temperature $T_{s0}$ is known. The reflectivity of the emitting layer cannot be obtained by any "juggling" from a single value of $Tb_p(f)$ unless $T_{s0}$ is specified.

4. The purpose of our article is to find such an estimate of reflectivity that would correlate with the thawed or frozen state of the soil (which we characterize by soil temperature). Since, in fact, it is not possible to estimate the effective temperature $T_{s0}$ for different landscapes, we equate it to $T_{s0}=Tb_V(6.9)$. Then we can offer some quantity, which is formally expressed by the formula (3)

$$\Gamma_p(f) = 1 - Tb_p(f)/T_{s0}. \tag{3}$$

to characterize the state of the emitting layer with effective temperature $Tb_V(6.9)$. Formally, from formula (3), $\Gamma_p(f)$ is the reflectivity. But, since $T_{s0}$ is not defined exactly, we have the right to characterize the value of $\Gamma_p(f)$ as an *apparent* value. It was possible to stop here and compare the time series of $Tb_p(f)$ and $\Gamma_p(f)$ or their gradients with the air temperature determined from the LST MODIS data. Because we do not derive a strictly mathematical parameter, but solve the problem by semi-empirically. But in this case, we could claim that we determine the state of only the visible interface between air and vegetation or air and snow. We couldn't say anything about the soil.

As it is known, observations of brightness temperature in the frequency range of 6.9-18.7 GHz were used to soil moisture retrieval. For example, in the model of microwave emission, which is used to retrieve soil moisture based on Nimbus-7 Scanning Multichannel Microwave Radiometer (SMMR) data, the calculation of brightness temperature of the soil covered with vegetation in the frequency range of 6.9-18.7 GHz is performed under the assumption that the temperature vegetation equals soil surface temperature (Njoku and Li, 1999).

Analogically, but and in contrast to the work (Njoku and Li, 1999), in accordance with formula (3), in our case, the effective temperature $T_{s0}$ of some emitting layer is used. In our work, we assume that all information about the effective temperature of some emitting layer $T_{s0}$ is contained in the value $Tb_V(6,9)$ measured by the satellite. Previously, we found that $Tb_V$ (6.9) better than others Tbs values in the frequency range of 6.9–18.7 GHz, correlates with the surface temperature of the tundra soil (Muzalevskiy et al., 2016). Therefore, from $Tb_H(f)$ (see equation (3) in manuscript), we can estimate soil reflectivity (depending on properties of snow and vegetation covers through effective soil roughness parameter $H(\theta,f)=Hr(\theta,f)+\tau(f)/\cos\theta$) as seen from equation (3B) in response to comment 7):

$$\Gamma_p(f) = 1 - Tb_p(f)/T_{s0}. \tag{3}$$

5. On Fig. 1 shows the time course of $Tb_H(f)$ and $\Gamma_H(f)$. The value of $\Gamma_H(f)$ was calculated based on formula (3) using the data of $Tb_H(f)$ and $Tb_V(6.9)$. $Tb_p(f)$ and $\Gamma_p(f)$ are different physical quantities. $Tb_p(f)$ is directly proportional to the soil temperature gradient according to formula (1) in manuscript, and $\Gamma_p(f)$ depends nonlinearly through the dielectric constant. The fact that Vasiliy Tikhonov (PhD) sees the result of the "recalculation" in the presented beautiful and synchronous graphs indicates the correctness of our assumption that reflectivity can be estimated based on a simple formula (3) using $Tb_V(6,9)$ values as an estimate of the soil surface temperature.

Based on the position of Vasiliy Tikhonov (PhD) in comments No. 4 and No. 6, our approach is completely wrong to perform such assessments and non-physical results will be obtained, and our approach deserves only such assessments by Vasiliy Tikhonov (PhD): «it is wrong to arbitrarily assume the 55-degree angle to be the soil Brewster angle» «To whom, how and why is it apparent value of reflectivity» «plenty of trivial or mutually exclusive statements throughout the text» «. The analysis of the satellite data is done by juggling the values» «The suggested physical interpretations also leave much to be desired».

Based on formula (2), the spectral gradient of the brightness temperature depends on the spectral gradient of reflectivity. The spectral gradient of reflectivity is independent of the spectral gradient of brightness temperature. From formula (2) it follows that in practice, it is impossible to "simply"

recalculate $\frac{\Delta Tb_p(f)}{\Delta f}$ into $\frac{\Delta \Gamma_p(f)}{\Delta f}$, since the value of the heat flux J0, either skin layer thickness, nor $\Gamma$p(f) are not known.

**References**

Njoku E.G. and Li Li, "Retrieval of land surface parameters using passive microwave measurements at 6-18 GHz," in IEEE Transactions on Geoscience and Remote Sensing, vol. 37, no. 1, pp. 79-93, Jan. 1999, doi: 10.1109/36.739125.

Muzalevskiy, K. V., Ruzicka, Z., Kosolapova, L. G., Mironov, V. L.: Temperature dependence of SMOS/MIRAS, GCOM-W1/AMSR2 brightness temperature and ALOS/PALSAR radar backscattering at arctic test sites, Proceedings of Progress in Electromagnetic Research Symposium (PIERS), Shanghai, 3578-3582, doi: 10.1109/PIERS.2016.7735375, 2016.

**Comment 9.** When considering the gradients of brightness temperature spectral densities and reflectivity "per unit interval of the frequency spectrum", Authors find that they "seem to be larger for the narrower 1.4-6.98 GHz than for the broader 1.4-36.5 GHz frequency band." There can be no doubt about this, since the discussed characteristics are obtained by dividing by a smaller value (frequency interval) in the first case and a larger value in the second case.

**Response to comment 9.** Vasiliy Tikhonov (PhD) statement would be correct if the brightness temperatures at different frequencies did not change. First, division by the corresponding frequency interval of the difference between the measured brightness temperatures it brings to a normalized value per unit frequency. Secondly, Vasiliy Tikhonov (PhD) do not take into account the significant variations in the brightness temperatures themselves depending on the frequency. Therefore, Vasiliy Tikhonov (PhD) conclusion is not obvious, especially in winter (see Fig. 1a in our manuscript).

**Comment 10.** The article is carelessly formatted. There are a number of typos both in the text and in the figure captions. In Figure 4, curve 1 merges in color with curve Ts0.

**Response to comment 10.** There is no figure with No. 4 in our manuscript. The comment does not apply to the article submitted to The Cryosphere.

**Comment 11.** In Introduction (p. 2), when considering various algorithms for determining the thawed and frozen soil states, the Authors mention the polarization index PR as an indicator. In the text, it is said that "The decision on thawed or frozen state of the soil is made when the normalized PR passes through 0." Based on the expression for PR, it should always be higher than 0, because, for any frequency, the

value of brightness temperature on vertical polarization is more than on horizontal. What then does the phrase "…when the normalized PR passes through 0" mean?

**Response to comment 11.** In accordance with the methodology described in the cited literature (Rautiainen et al., 2014), "normalized" means the following:

$$FF_{rel\_X}(t) = \frac{(FF_X(t) - FF_{SUMMER\_X})}{(FF_{WINTER\_X} - FF_{SUMMER\_X})} \cdot 100 \qquad (3)$$

:

where $FF_X$ can be equal PR.

**Comment 12.** The last sentence of Introduction concludes: "Taking into account the development of domestic multifrequency satellite radiometric sensing systems and the expected launch in 2028 of the multispectral (1.4-36.5 GHz) Copernicus Imaging Microwave Radiometer (Kilic et al, 2018) of high spatial resolution (55-5 km), development of new multifrequency radiometric methods to identify thawed/frozen soil state is highly relevant". I wonder, what are these "domestic radiometric multifrequency satellite sensing systems" and why only Copernicus is given as an example, and not some domestic system?

**Response to comment 12.** Currently, there are several Russian satellites of the Meteor series in orbit (Mitnik et al., 2017; https://space.oscar.wmo.int/instruments/view/mtvza_gy), these satellites are equipped with a multispectral radiometer MTVZA-GA with a frequency range of 10.6-183.3GHz.

**Reference**

*Mitnik L., Kuleshov V., Mitnik M., Streltsov A.M., Cherniavsky G., Cherny I.* Microwave scanner sounder MTVZA-GY on new Russian meteorological satellite Meteor-M N 2: modeling, calibration and measurements // IEEE Journal of Selected Topics in Applied Earth Observations and Remote Sensing. 2017. Vol. 10. N. 7. P. 3036-3045.

**References**

1. Bircher, S.; Demontoux, F.; Razafindratsima, S.; Zakharova, E.; Drusch, M.; Wigneron, J.-P.; Kerr, Y.H. L-Band Relative Permittivity of Organic Soil Surface Layers—A New Dataset of Resonant Cavity Measurements and Model Evaluation. Remote Sens. 2016, 8, 1024. https://doi.org/10.3390/rs8121024

2. Bogorodsky V.V., Kozlov A.I. Mikrovolnovaya radiometriya zemnyh pokrovov. [Microwave radiometry of earth covers]. Gidrometeoizdat. Leningrad. 1985. 272p.

3. Bogorodsky V.V., Kozlov A.I., Tuchkov L.T. Radioteplovoe izluchenie zemnyh pokrovov. [Radiothermal radiation of the earth's covers.] Gidrometeoizdat. Leningrad. 1977. 224p.

4. Brekhovskikh L.M. Waves in Layered Media, NewYork, NY, USA: Academic. 1960. P. 561.

5. CALM_Data. Available online, 2022: https://www2.gwu.edu/~calm/data/CALM_Data/North%20America/Alaska/North%20Slope/U12_toolik/

6. Calvet, J.-C., J.-P. Wigneron, E. Mougin, Y. H. Kerr, and J. S. Brito (1994): Plant water content and temperature of the Amazon forest from satellite microwave radiometry. IEEE Trans. Geosci. Rem. Sens., 32, 397-408.

7. Dostovalov B.N., Kudryavtsev V.A. Obshchee merzlotovenie [Permafrost. General physical basics]. Publishing House of Moscow State University, Moscow, 1967. 404p. (In Russian)

8. Duan, S.-B.; Han, X.-J.; Huang, C.; Li, Z.-L.; Wu, H.; Qian, Y.; Gao, M.; Leng, P. Land Surface Temperature Retrieval from Passive Microwave Satellite Observations: State-of-the-Art and Future Directions. Remote Sens. 2020, 12, 2573. https://doi.org/10.3390/rs12162573

9. Gao L, Qiang Gao, Hankui Zhang, Xiaojun Li, Mario Julian Chaubell, Ardeshir Ebtehaj, Lian Shen, Jean-Pierre Wigneron, A deep neural network based SMAP soil moisture product, Remote Sensing of Environment, Volume 277, 2022, 113059.

10. Hachem, S., Duguay, C. R., and Allard, M.: Comparison of MODIS-derived land surface temperatures with ground surface and air temperature measurements in continuous permafrost terrain, The Cryosphere, 6, 51–69, https://doi.org/10.5194/tc-6-51-2012, 2012.

11. Harris S.A., H.M. French, J.A. Heginbottom, G.H. Johnston, B. Ladanyi, D.C. Sego, R.O. van Everdingen National Research Council of Canada Ottawa, Ontario, Canada KIA OR6. 1988.

12. He H., Dyck M. Application of Multiphase Dielectric Mixing Models for Understanding the Effective Dielectric Permittivity of Frozen Soils.Vadose Zone Journal. 2013. 10.2136/vzj2012.0060

13. Ikonen, J., Vehviläinen, J., Rautiainen, K., Smolander, T., Lemmetyinen, J., Bircher, S., and Pulliainen, J.: The Sodankylä in situ soil moisture observation network: an example application of ESA CCI soil moisture product evaluation, Geosci. Instrum. Method. Data Syst., 5, 95–108, https://doi.org/10.5194/gi-5-95-2016, 2016.

14. Ikonen, Jaakko, Tuomo Smolander, Kimmo Rautiainen, Juval Cohen, Juha Lemmetyinen, Miia Salminen, and Jouni Pulliainen. 2018. "Spatially Distributed Evaluation of ESA CCI Soil Moisture Products in a Northern Boreal Forest Environment" Geosciences 8, no. 2: 51. https://doi.org/10.3390/geosciences8020051

15. Jones L. A., J. S. Kimball, K. C. McDonald, S. T. K. Chan, E. G. Njoku and W. C. Oechel, "Satellite Microwave Remote Sensing of Boreal and Arctic Soil Temperatures From AMSR-E," in IEEE Transactions on Geoscience and Remote Sensing, vol. 45, no. 7, pp. 2004-2018, July 2007, doi: 10.1109/TGRS.2007.898436.

16. Lara, R. P., Berg, A. A., Warland, J., Tetlock, E. (2020). In situ estimates of freezing/melting point depression in agricultural soils using permittivity and temperature measurements. Water Resources Research, 56, e2019WR026020. https://doi.org/10.1029/2019WR026020

17. Lemmetyinen J., Mike Schwank, Kimmo Rautiainen, Anna Kontu, Tiina Parkkinen, Christian Mätzler, Andreas Wiesmann, Urs Wegmüller, Chris Derksen, Peter Toose, Alexandre Roy, Jouni Pulliainen, Snow density and ground permittivity retrieved from L-band radiometry: Application to experimental data, Remote Sensing of Environment, Volume 180, 2016, Pages 377-391.

18. Lukin Yu. I., V. L. Mironov, S. A. Komarov. Investigation of the dielectric spectra of wet soil during freezing-thawing // Izvestiya vuzov. Physica. — 2008. — Vol. 51. No. 9. —P. 24–28.

19. Mavrovic A., Renato Pardo Lara, Aaron Berg, François Demontoux, Alain Royer, Alexandre Roy. Soil dielectric characterization during freeze–thaw transitions using L-band coaxial and soil moisture probes. Hydrol. Earth Syst. Sci., 25, 1117–1131, 2021. https://doi.org/10.5194/hess-25-1117-2021

20. McFarland, M. J., R. L. Miller, and C. M. U. Neale (1990): Land surface temperature derived from the SSM/I passive microwave brightness temperatures. IEEE Trans. Geosci. Rem. Sens., 28(5), 839-845.

21. Mialon A. et al., "Comparison of Dobson and Mironov Dielectric Models in the SMOS Soil Moisture Retrieval Algorithm," in IEEE Transactions on Geoscience and Remote Sensing, vol. 53, no. 6, pp. 3084-3094, June 2015, doi: 10.1109/TGRS.2014.2368585.

22. Mironov V.L., A.Yu Karavayskiy, Yu.I. Lukin, Pogoreltsev E.I. Joint studies of water phase transitions in Na-bentonite clay by calorimetric and dielectric methods. Cold Regions Science and Technology. 2018. Vol. 153. P. 172-180.

23.	Mironov et al. Temperature- and texture-dependent dielectric model for frozen and thawed mineral soils at a frequency of 1.4 GHz. REMOTE SENSING OF ENVIRONMENT. 2017. Vol. 200. P. 240-249; 10.1016/j.rse.2017.08.007

24.	Mironov V. L., L. G. Kosolapova, S. V. Fomin and I. V. Savin, "Experimental Analysis and Empirical Model of the Complex Permittivity of Five Organic Soils at 1.4 GHz in the Temperature Range From −30 °C to 25 °C," in IEEE Transactions on Geoscience and Remote Sensing, vol. 57, no. 6, pp. 3778-3787, June 2019, doi: 10.1109/TGRS.2018.2887117.

25.	Mironov V., Y. Kerr, J. Wigneron, L. Kosolapova and F. Demontoux, "Temperature- and Texture-Dependent Dielectric Model for Moist Soils at 1.4 GHz," in IEEE Geoscience and Remote Sensing Letters, vol. 10, no. 3, pp. 419-423, May 2013, doi: 10.1109/LGRS.2012.2207878.

26.	Mironov V.L., Muzalevskiy K.V. Measurement method of the temperature dependence of the dielectric permittivity of topsoil on the Yamal Peninsula using data of the radiometer MIRAS SMOS// Regional problems of remote sensing of the Earth. 2014. C. 186-189. (in Russian). Available online: http://rprs.sfu-kras.ru/sites/default/files/v_pechat_-materialy_mezhdunarodnoy_nauchnoy_konferencii.pdf (p. 186-189)

27.	Mironov V. L., R. D. De Roo and I. V. Savin, "Temperature-Dependable Microwave Dielectric Model for an Arctic Soil," in IEEE Transactions on Geoscience and Remote Sensing, vol. 48, no. 6, pp. 2544-2556, June 2010, doi: 10.1109/TGRS.2010.2040034

28.	Mitnik L., Kuleshov V., Mitnik M., Streltsov A.M., Cherniavsky G., Cherny I. Microwave scanner sounder MTVZA-GY on new Russian meteorological satellite Meteor-M N 2: modeling, calibration and measurements // IEEE Journal of Selected Topics in Applied Earth Observations and Remote Sensing. 2017. Vol. 10. N. 7. P. 3036-3045.

29.	Muskett, R. , Romanovsky, V. , Cable, W. and Kholodov, A. (2015) Active-Layer Soil Moisture Content Regional Variations in Alaska and Russia by Ground-Based and Satellite-Based Methods, 2002 through 2014. International Journal of Geosciences, 6, 12-41. doi: 10.4236/ijg.2015.61002.

30.	Muzalevskiy K., Ruzicka Z., Roy A., Loranty M., Vasiliev A. Classification of the frozen/thawed surface state of Northern land areas based on SMAP and GCOM-W1 brightness temperature observations at 1.4 GHz and 6.9 GHz. Remote Sensing Letters. 2021. Vol. 12. No. 11. P. 1073-1081. 10.1080/2150704X.2021.1963497

31.	Muzalevskiy K., Ruzicka Z. Detection of soil freeze/thaw states in the Arctic region based on combined SMAP and AMSR-2 radio brightness observations // International Journal of Remote Sensing. – 2020. – V. 41.- Is. 14. – P. 5046-5061. DOI: 10.1080/01431161.2020.1724348.

32.	Muzalevskiy K. V. and Z. Ruzicka, "Retrieving Soil Temperature at a Test Site on the Yamal Peninsula Based on the SMOS Brightness Temperature Observations," in IEEE Journal of Selected Topics in Applied Earth Observations and Remote Sensing, vol. 9, no. 6, pp. 2468-2477, June 2016, doi: 10.1109/JSTARS.2016.2553220.

33.	Muzalevskiy, K. V., Ruzicka, Z., Kosolapova, L. G., Mironov, V. L.: Temperature dependence of SMOS/MIRAS, GCOM-W1/AMSR2 brightness temperature and ALOS/PALSAR radar backscattering at arctic test sites, Proceedings of Progress in Electromagnetic Research Symposium (PIERS), Shanghai, 3578-3582, doi: 10.1109/PIERS.2016.7735375, 2016.

34.	Njoku E.G. and Li Li, "Retrieval of land surface parameters using passive microwave measurements at 6-18 GHz," in IEEE Transactions on Geoscience and Remote Sensing, vol. 37, no. 1, pp. 79-93, Jan. 1999, doi: 10.1109/36.739125.

35.	Parrens M., Jean-Pierre Wigneron, Philippe Richaume, Ahmad Al Bitar, Arnaud Mialon, Roberto Fernandez-Moran, Amen Al-Yaari, Peggy O'Neill, Yann Kerr, Considering combined or separated roughness and vegetation effects in soil moisture retrievals, International Journal of Applied Earth Observation and Geoinformation, Volume 55, 2017, Pages 73-86.

36.	Pulliainen, J. T., J. Grandell, and M. T. Hallikainen (1997): Retrieval of surface temperature in Boreal forest zone from SSM/I data. IEEE Trans. Geosci. Rem. Sens., 35, 1188-1200.

37.	RAO K. S., GIRISH CHANDRA & P. V. NARASIMHA RAO (1987) The relationship between brightness temperature and soil moisture Selection of frequency range for microwave remote sensing, International Journal of Remote Sensing, 8:10, 1531-1545, DOI:10.1080/01431168708954795

38.	Rodriguez-Alvarez N. et al., "Land Geophysical Parameters Retrieval Using the Interference Pattern GNSS-R Technique," in IEEE Transactions on Geoscience and Remote Sensing, vol. 49, no. 1, pp. 71-84, Jan. 2011, doi: 10.1109/TGRS.2010.2049023.

39.	Roy A.; Leduc-Leballeur, M.; Picard, G.; Royer, A.; Toose, P.; Derksen, C.; Lemmetyinen, J.; Berg, A.; Rowlandson, T.; Schwank, M. Modelling the L-Band Snow-Covered Surface Emission in a Winter Canadian Prairie Environment. Remote Sens. 2018, 10, 1451. https://doi.org/10.3390/rs10091451

40.	Santi, E.; Pettinato, S.; Paloscia, S.; Pampaloni, P.; Fontanelli, G.; Crepaz, A.; Valt, M. Monitoring of Alpine snow using satellite radiometers and artificial neural networks. Remote Sensing of Environment. 2014. 10.1016/j.rse.2014.01.012

41.	Schwank M. et al. Model for microwave emission of a snow-covered ground with focus on L band, Remote Sensing of Environment, Volume 154, 2014, Pages 180-191.

42.	Sirounian, V. The effect of the temperature, angle of observation, salinity, and thin ice on the microwave emission of water. J. Geophys. Res., 1968, Vol. 73, No. 14, p. 4481-4486.

43.	Schmugge T.J. Soil moisture measurements with microwave radiometers. Report. NASA. No. x-652-72-305. 1972

44.     Tao, J., Koster, R. D., Reichle, R. H., Forman, B. A., Xue, Y., Chen, R. H., and Moghaddam, M.: Permafrost variability over the Northern Hemisphere based on the MERRA-2 reanalysis, The Cryosphere, 13, 2087–2110, https://doi.org/10.5194/tc-13-2087-2019, 2019.

45.     "The Authoritative Dictionary of IEEE Standards Terms, Seventh Edition," in IEEE Std 100-2000 , vol., no., pp.1-1362, 11 Dec. 2000, doi: 10.1109/IEEESTD.2000.322230.).

46.     Ulaby, F. T., Long, D. G., Blackwell, W. J., Elachi, C., Fung, A. K., Ruf, C., Sarabandi, K., Zebker, H. A., and Van Zyl, J. Microwave radar and radiometric remote sensing, University of Michigan Press Ann Arbor, MI, USA, 2014.

47.     Wigneron J.-P., T.J. Jackson, P. O'Neill, G. De Lannoy, P. de Rosnay, J.P. Walker, P. Ferrazzoli, V. Mironov, S. Bircher, J.P. Grant, M. Kurum, M. Schwank, J. Munoz-Sabater, N. Das, A. Royer, A. Al-Yaari, A. Al Bitar, R. Fernandez-Moran, H. Lawrence, A. Mialon, M. Parrens, P. Richaume, S. Delwart, Y. Kerr, Modelling the passive microwave signature from land surfaces: A review of recent results and application to the L-band SMOS & SMAP soil moisture retrieval algorithms, Remote Sensing of Environment, Volume 192, 2017, Pages 238-262.

48.     Wigneron J. -P. et al., "Evaluating an Improved Parameterization of the Soil Emission in L-MEB," in IEEE Transactions on Geoscience and Remote Sensing, vol. 49, no. 4, pp. 1177-1189, April 2011, doi: 10.1109/TGRS.2010.2075935.

49.     Zhang, L.; Shi, J.; Zhang, Z.; Zhao, K. The estimation of dielectric constant of frozen soil-water mixture at microwave bands. In Proceedings of the 2003 IEEE International Geoscience and Remote Sensing Symposium, Toulouse, France, 21–25 July 2003; Volume 4, pp. 2903–290.

50.     Zuerndorfer, B., England, A. W., Radiobrightness decision criteria for freeze/thaw boundaries, IEEE Transactions on Geoscience and Remote Sensing, 30, 1, 89-102, doi:10.1109/36.124219, 1992.

---

## Author Comment (AC3)

**Response to RC2: 'Comment on egusphere-2022-224', Anonymous Referee #2, 18 Nov 2022.**
**https://doi.org/10.5194/egusphere-2022-224-RC2**

**Comment 1.** In my opinion this manuscript communicates an interesting and potentially important observation that the spectral gradients of brightness temperatures can be used to detect the freezing / thawing of soil. I also think that the paper is well written.

**Response to comment 1.** Thank you very much, for your opinion and support. You and RC1 feedback gave me back the ground under my feet and the possibility of further work on the article.

**Comment 2.** Line 30: the authors say that the single frequency methods use viewing angle of about 40 degrees. At least in case of SMOS also multiangular data is available even though the authors mentioned in the introduction might not have used it.

**Response to comment 2.** Yes, we meant this possibility. But we chose the SMAP data. SMAP is the real aperture radiometer (Spencer et al, 2010). SMOS is the interferometric radiometer (Kerr et al, 2000). In addition, each brightness temperature in the range of viewing angles from 10 to 65 degrees is measured by SMOS at different azimuth angles (see Fig. 1). These two factors cause, that the SMOS brightness temperature to be measured with additional fluctuations, which are a source of additional random error with respect to the SMAP data. As you can see (Fig. 2, Fig. 3), such fluctuations in the brightness temperature of SMOS are almost twice as large as those of SMAP. That is why we chose SMAP data.

[Figure]

[Figure]

Fig. 1. Azimuth angle vs incidence angle at the central part of the Great Bear Lake

[Figure]

[Figure]

Fig. 2. Time series of brightness temperature (H-pol), measured by SMOS and SMAP at the viewing angle of approximately 40 degree over central part of the Great Bear lake.

Fig. 3. Distribution of brightness temperature (H-pol) over central part of the Great Bear lake (based on data in Fig. 2).

**References**

Kerr Y. et al., "New radiometers: SMOS-a dual pol L-band 2D aperture synthesis radiometer," 2000 IEEE Aerospace Conference. Proceedings (Cat. No.00TH8484), 2000, pp. 119-128 vol.5, doi: 10.1109/AERO.2000.878481.

Spencer M. et al., "The Soil Moisture Active Passive (SMAP) mission L-Band radar/radiometer instrument," 2010 IEEE International Geoscience and Remote Sensing Symposium, 2010, pp. 3240-3243, doi: 10.1109/IGARSS.2010.5651760.

**Comment 3.** Lines 133-134, Fig. 1: the ovals mentioned in the text are practically invisible in Fig. 1 and their visibility must be improved.

**Response to comment 3.** The figures have been replaced. In the new version of the manuscript the selected areas are shown as rectangles with a dash line. In the new version of the manuscript word "ovals" was replaced to "rectangles" on

Line 134: "Several such regions are marked by rectangles with a dashed line in Fig. 1a, 1b (see the period 2015-2016)."

[Figure]

Fig. 4. (in manuscript Fig. 1.)

**Comment 4.** Fig. 2: the color of Ts0 and first gradient pair is very hard to distinguish. A different set of colors or line types should be applied.

**Response to comment 4.** In the new version of the manuscript, I changed the color of the Ts0 line to orange.

[Figure]

Fig 5. (in manuscript Fig. 2.)

**Comment 5.** Line 175: If the soil is dry the penetration depth at 1.4 GHz is couple of centimeters. Have the authors considered that the difference between using 6.9-1.4 GHz and 36.5-1.4 GHz could be related to this?

**Response to comment 5.** I didn't fully understand the question, but I'll try to answer. When the soil is dry, the thickness of emitting layer is greater than when the soil is wet. For wet soil, the emissivity in the L-band is mainly formed by the topsoil of 0-2 cm (Schmugge, 1980, Escorihuela et al, 2010). In the case of dry soil, the depth will increase slightly. By itself, the observation of the brightness temperature at horizontal polarization and a frequency of 1.4 GHz does not make it possible to unambiguously identify a thawed or frozen state (see Fig. 6 below). In this regard, we have proposed an approach that allows us to obtain a time series in Fig. 5 (in

manuscript Fig. 2.) from the time series in Fig. 6 (below), which has a significantly greater unambiguity. From our point of view, namely the gradient of the spectral density brightness temperature or reflectivity demonstrates the difference in emissivity of the layers placed at different depths in inhomogeneous structure of the snow-vegetation-soil cover.

[Figure]

Fig. 6. Brightness temperature at H-pol (1.4GHz). HV test site.

**References**

Escorihuela M.J. et al. Effective soil moisture sampling depth of L-band radiometry: A case study. Remote Sensing of Environment, Elsevier, 2010, 114 (RSE-07569), pp.7. ff10.1016/j.rse.2009.12.011ff. ffird-00467991ff

Schmugge T. J. Effect of soil texture on the microwave emission from soils. 1980. NASA-TM-80632. 32 p. (https://ntrs.nasa.gov/citations/19800010256)

---

## Author Comment (AC4)

**Response to RC1: 'Comment on egusphere-2022-224', Anonymous Referee #1, 16 Sep 2022.**
**https://doi.org/10.5194/egusphere-2022-224-RC1**

**Comment 0.** I believe this is an interesting manuscript, which will contribute to the literature on freeze/thaw classification. However, I do believe revisions are necessary before publication. I noted some more detailed points below, but in summary I believe that some additional discussions on the frequencies/angles used, and the inclusion of the other study sites as well as some discussion on the differences between the sites and the impact of this on the results would be critical.

**Response to comment 0.** Thank you very much, for your opinion and support. You and RC2 feedbacks gave me back the ground under my feet and the possibility of further work on the article.

Major comments:

**Comment 1.**

A) Line 122: "The physical basis for this effect is the observation angle of the AMSR-2 radiometer of GCOM-W1 satellite, which is close to the Brewster angle (55º)": please provide a citation(s) for this paragraph.

B) Line 98: "The difference in observation angles 40° and 55°, respectively for SMAP and AMSR2, was neglected.": why was this neglected, and what is the justification. As far as I could tell, there is no discussion on this in section 4. I believe a thorough discussion of this is necessary as this is at the basis of this methodology.

**Response to comment 1.**

A)      AMSR2 viewing angle of 55° is very close to Brewster's angle for a natural rough surface or soil surface covered with vegetation or snow. This is not far from reality. The natural soil surface is always rough surface. If the roughness increases, Brewster's angle decreases (Ulaby, 2014, Fig. 10-17 and text on p. 437). The angle already reaches 57 degrees when roughness increases to 0.926 cm (see Fig. 10-17 below). The roughness of natural tundra soils has much higher values, which change in a wide range from 1.06 cm to 4.28 cm. (Watanabe et al., 2012).

[Figure]

**Figure 10-17:** Comparison of I²EM-computed bistatic scattering coefficient with measurements made in the incidence plane ($\theta_i = \theta_s$ and $\phi = 0$) for three surfaces with different roughnesses.

[Figure]

**Figure 12-5:** Measured brightness temperatures as a function of incidence angle at (a) 1.4 GHz, (b) 5 GHz, and (c) 10.7 GHz. The soil temperature for both smooth and rough fields was $\sim 20$ °C. The volumetric soil moisture content for the smooth field was $\sim 0.25$ g/cm³ and for the rough field was $\sim 0.26$ g/cm³ in the top 0 to 10 cm layer [Wang et al., 1983].

**Reference to figures.** Ulaby, F. T., Long, D. G., Blackwell, W. J., Elachi, C., Fung, A. K., Ruf, C., Sarabandi, K., Zebker, H. A., and Van Zyl, J. Microwave radar and radiometric remote sensing, University of Michigan Press Ann Arbor, MI, USA, 2014

     The natural soil is covered with vegetation. As a result of interference in the canopy, the region of Brewster's angle is blurred and may contain many minimums. This can be seen, for example in (Rodriguez-Alvarez, Fig. 7, 8). Brewster's angle for vegetated soil can even be as high as 40-50° (GNSS satellite elevation) or 50-40° (in viewing angle) (Rodriguez-Alvarez, 2011, Figs. 7, 8).

[Figure]

Fig. 7. Simpler model, vegetation-covered soils case. Simulated interference power received versus reflectivity. (a) Bare soil produces one notch, (b) 60-cm vegetation layer + soil layer produces three notches, and (c) 90-cm vegetation layer + soil layer produces four notches. Note that first notch is due to the Brewster's angle, but new notches appear due to oscillations in the reflectivity.

**Reference to Fig.7.** Rodriguez-Alvarez N. *et al.*, "Land Geophysical Parameters Retrieval Using the Interference Pattern GNSS-R Technique," in *IEEE Transactions on Geoscience and Remote Sensing*, vol. 49, no. 1, pp. 71-84, Jan. 2011, doi: 10.1109/TGRS.2010.2049023.

Also, experimentally measured values of brightness temperature over thawed or frozen tundra soil (covered with snow) show that the maximum brightness temperature on vertical polarization is observed in the range of angles 45-70° (taking into account the measurement error of satellite radiometers) (Lemmetyinen et al, 2016, Fig.5-7; Ulaby et al, 2014, Fig. 12-40).

Brewster's angle is 55-65°          Brewster's angle is 45-55°

[Figure]

Fig. 5. Simulated (lines) and measured (symbols) L-band brightness temperatures $T_B^p(\theta_k)$ ($p$ = H, V) as function of the incidence angle $\theta_k$ (solid lines: horizontal polarization ($p$ = H); dashed lines: vertical polarization ($p$ = V). Simulations using the indicated snow densities $\rho_S$ are shown in comparison with tower-based ELBARA-II observations (triangles) performed at the forest clearing site. All simulations were made for the indicated ground permittivity $\varepsilon_G$ and ground temperature $T_G$ measured from SO11 sensors at 5 cm depth.

[Figure]

Fig. 6. Simulated (lines and shaded areas) and measured (triangles) $T_B^p(\theta_k)$ for the snow-covered footprints at the wetland site and for the corresponding footprints after snow clearing. Extent of snow clearance indicated with vertical dashed lines.

Brewster's angle are a) 60-70°, b) 55-70°, c) 55-65°, d) 55-65°

[Figure]

**Fig. 7.** Examples of simulated (lines) and observed (triangles) brightness temperatures $T_B^p(\theta_k)$ using the indicated retrieved parameters $\mathbf{P} = (\varepsilon_{G,ret}, \rho_{S,ret})$. Forest clearing site for summer (a) and winter (b); wetland site for late autumn (c) and winter (d). Observation angle ranges of $40° \leq \theta_k \leq 60°$ and $50° \leq \theta_k \leq 60°$ used for retrievals over the forest clearing and wetland sites, respectively. Hollow triangles indicate snow free conditions, filled triangles indicate presence of snow.

**Reference to Fig. 7.** Lemmetyinen J., Mike Schwank, Kimmo Rautiainen, Anna Kontu, Tiina Parkkinen, Christian Mätzler, Andreas Wiesmann, Urs Wegmüller, Chris Derksen, Peter Toose, Alexandre Roy, Jouni Pulliainen, Snow density and ground permittivity retrieved from L-band radiometry: Application to experimental data, Remote Sensing of Environment, Volume 180, 2016, Pages 377-391.

Brewster's angle are <40-60°

[Figure]

**Figure 12-40:** Comparison of theory with angular measurements for a snow layer with $\varepsilon_{ice} = 3.15 - j14 \times 10^{-3}$, snow density $\rho_s = 0.3$ g/cm$^3$, equivalent scatterer radius $r = 0.85$ mm, and wetness by weight $m_w = 0.06$ percent [Fung and Eom, 1981].

**Reference to Fig. 12-40.** Ulaby, F. T., Long, D. G., Blackwell, W. J., Elachi, C., Fung, A. K., Ruf, C., Sarabandi, K., Zebker, H. A., and Van Zyl, J. Microwave radar and radiometric remote sensing, University of Michigan Press Ann Arbor, MI, USA, 2014

Our calculations, which we additionally carried out for a homogeneous soil, covered with a layer of snow or vegetation, also confirm that Brewster's angle is blurred and is in the region of the AMSR2 viewing angle (see Fig. 1A, below). In the calculations, the content of organic matter in the soil was set equal to 50% (by weight), soil moisture 45% (by volume). The temperature of thawed and frozen soil was set equal to 20°C and -10°C, respectively. The soil permittivity was calculated at a frequency of 1.4 GHz based on formulas from (Mironov et al, 2019).

Reference. Mironov V. L., L. G. Kosolapova, S. V. Fomin and I. V. Savin, "Experimental Analysis and Empirical Model of the Complex Permittivity of Five Organic Soils at 1.4 GHz in the Temperature Range From −30 °C to 25 °C," in IEEE Transactions on Geoscience and Remote Sensing, vol. 57, no. 6, pp. 3778-3787, June 2019, doi: 10.1109/TGRS.2018.2887117.

[Figure]

**Figure 1A.** Modulus of reflection coefficient (V-pol) for soil covered with vegetation (red) and snow(blue) layer. The density of dry snow was set equal to 0.32g/cm$^3$, and the effective permittivity of the vegetation cover was taken to be 1.47+0.36i (Schwank, 2014). **Reference.** Schwank M. et al. Model for microwave emission of a snow-covered ground with focus on L band, Remote Sensing of Environment, Volume 154, 2014, Pages 180-191

We also note the following fact. The emissivity on vertical polarization in the range of angles slightly less than Brewster's angle, weakly depends on the properties of snow canopy over soil (Schwank et al., 2014, see Fig. 9). As the snow density increases, Brewster's angle decreases and tends from 65° to ~57–50°. It can be assumed that the above results and similar conclusions are also valid when soil covered with vegetation layer.

Reference. Schwank M. et al. Model for microwave emission of a snow-covered ground with focus on L band, Remote Sensing of Environment, Volume 154, 2014, Pages 180-191.

[Figure]

**Reference to Fig. 9.** Schwank M. et al. Model for microwave emission of a snow-covered ground with focus on L band, Remote Sensing of Environment, Volume 154, 2014, Pages 180-191. (Also see Fig. 10 in (Schwank et al, 2014).)

**Fig. 9.** Angular dependences of emissivities $E^p(\theta_0)$ of GS systems with frost-depth $d_{FG} =$ 40 cm. Definitions of the SPs are provided in Table 2.

The analysis performed convincingly shows that the brightness temperature measured by AMSR2 at an angle of 55° is very close to the Brewster angle for a natural rough surface or soil surface covered with vegetation or snow. In addition, it should be taken into account that soils of different

moisture content, height and biomass of vegetation, different height and density of snow fall into the ~50x50 km pixel, which leads to even more blurring and smoothing of the Brewster angle area.

B)       We proposed a semi-empirical approach to identify thawed and frozen states of soils. And we don't have requirements to follow mathematical rigor in formulas. We use the formulas as the main highways of radiation laws, in which, due to the empirical approach, we use brightness temperatures that do not exactly match in the sensing angle. In any case, the neglect in the sensing angle that we allowed is contained in the total error of the proposed method.

However, let us try to substantiate our assumption a little more rigorously. In accordance with expression (3) from the manuscript, when calculating

$$\Gamma_p(\theta=40°)=1-Tb_H(\theta=40°)/ Tb_V(\theta=55°),$$

the observation angles do not coincide. $Tb_H(\theta=40°)$ measured by SMAP at 40° and $Tb_V(\theta=55°)$ measured by AMSR2/GCOM-1 at 55°. Please pay attention to emission model of **bare** soil (Wigneron et al., 2011), whose parameters were found on the basis of experimental data. Accuracy of the model do not better than ±4–5 K.

Reference. Wigneron J.-P., Chanzy, A., Kerr, Y., Lawrence, H. et al.: Evaluating an Improved Parameterization of the Soil Emission in L-MEB, IEEE Transactions on Geoscience and Remote Sensing, 49, 4, 1177-1189, doi: 10.1109/TGRS.2010.2075935, 2011.

In reality, soil covered with snow and vegetation within a footprint area of tens km, this error should increase significantly higher than 4-5K. Let us turn to the experimentally measured angular values of brightness temperature on horizontal and vertical polarizations over frozen and thawed tundra soil (covered and not covered with snow) (Lemmetyinen et al, 2016, Fig. 7) and soil in Canadian Prairie (Roy et al, 2018, Fig. 3)..

[Figure]

**Fig. 7.** Examples of simulated (lines) and observed (triangles) brightness temperatures $T_B^p(\theta_k)$ using the indicated retrieved parameters $\mathbf{P} = (\varepsilon_{G,ret}, \rho_{S,ret})$. Forest clearing site for summer (a) and winter (b); wetland site for late autumn (c) and winter (d). Observation angle ranges of $40° \leq \theta_k \leq 60°$ and $50° \leq \theta_k \leq 60°$ used for retrievals over the forest clearing and wetland sites, respectively. Hollow triangles indicate snow free conditions, filled triangles indicate presence of snow.

**Reference to Fig. 7.** Lemmetyinen J., Mike Schwank, Kimmo Rautiainen, Anna Kontu, Tiina Parkkinen, Christian Mätzler, Andreas Wiesmann, Urs Wegmüller, Chris Derksen, Peter Toose, Alexandre Roy, Jouni Pulliainen, Snow density and ground permittivity retrieved from L-band radiometry: Application to experimental data, Remote Sensing of Environment, Volume 180, 2016, Pages 377-391.

[Figure]

**Figure 3.** $T_B$V-pol (blue) and $T_B$H-pol (black) on 4 March 2015 at Scene 1 measured (symbols) and simulated (lines) with the Wave Approach for LOw-frequency MIcrowave emission in Snow (WALOMIS) with added Gaussian noise ($\sigma_d$) to the measured density of 30 kg m$^{-3}$ (**left**), 60 kg m$^{-3}$ (**center**) and 90 kg m$^{-3}$ (**right**).

**Reference to Fig. 3.** Roy A.; Leduc-Leballeur, M.; Picard, G.; Royer, A.; Toose, P.; Derksen, C.; Lemmetyinen, J.; Berg, A.; Rowlandson, T.; Schwank, M. Modelling the L-Band Snow-Covered Surface Emission in a Winter Canadian Prairie Environment. *Remote Sens.* **2018**, *10*, 1451. https://doi.org/10.3390/rs10091451

It can be seen from the figures above that, within the error of ±5K, the angular dependence of the brightness temperature on the H-pol can be neglected within the variation of angles of 40-55°. (The exception is flooded soil, Fig. 7c). As a result, formula (3) from the manuscript can be used within the errors of measurements ±1.3-1.5 K (Piepmeier et al., 2017; Gao et al., 2019) and emission model ±4-5K (Wigneron et al., 2011)

**References.** Gao, S., Li, Z., Chen, Q., Zhou, W., Lin, M., Yin, X.: Inter-Sensor Calibration between HY-2B and AMSR2 Passive Microwave Data in Land Surface and First Result for Snow Water Equivalent Retrieval, Sensors, 19, 5023, doi:10.3390/s19225023, 2019.

Piepmeier, J.R., Focardi, P., Horgan, K.A., Knuble, J. et al.: SMAP L-Band Microwave Radiometer: Instrument Design and First Year on Orbit, IEEE Trans Geosci Remote Sens. 2017, 55, 4, 1954-1966, doi: 10.1109/tgrs.2016.2631978, 2017.

Wigneron J.-P., Chanzy, A., Kerr, Y., Lawrence, H. et al.: Evaluating an Improved Parameterization of the Soil Emission in L-MEB, IEEE Transactions on Geoscience and Remote Sensing, 49, 4, 1177-1189, doi: 10.1109/TGRS.2010.2075935, 2011.

In accordance with your comments in the new version of the manuscript, additional explanations are made after equation (3):

" Indeed, reflection coefficient measurements show that as root-mean-square (RMS) heights of soil surface roughness increases from 0.25 cm to 0.93 cm, Brewster's angle decreases from 60° to 57°. (De Roo and Ulaby, 1994, also see Wang et al., 1983, Fig. 2). The roughness of natural tundra soils has much higher values, which change in a wide range from 1.06 cm to 4.28 cm (Watanabe et al., 2012). The presence of vegetation (snow) cover on a rough soil surface leads to blurring and flattening of the V-pol angular dependence of reflectivity (Rodriguez-Alvarez et al., 2011, see Figs. 7, 8) and brightness temperature (Lemmetyinen et al., 2016, see Figs. 5-7; Chang and Shiue, 1980, see Fig. 3-5) in the region of Brewster's angles, due to the interference phenomenon. Within the error measurement of brightness temperature of

±1.3–1.5 K (Piepmeier et al., 2017; Gao et al., 2019), as well as the accuracy of emission models of ±4–5 K (Wigneron et al., 2011), Brewster's angle can be determined within a wide range of about from 45° to 65° (Lemmetyinen et al., 2016, see Fig. 5-7; Chang and Shiue, 1980, see Fig. 3-5). The error measurement of brightness temperature and the accuracy of emission models also makes it possible to neglect the variations of H-pol brightness temperatures of snow(vegetation)-covered soil in a range of observation angles from 40° to 55° (Roy et al., 2018, see Fig. 3; Lemmetyinen et al., 2016, see Fig. 7; Chang and Shiue, 1980, see Fig. 3-5). In this regard, the use of brightness temperatures measured at different angles in equation (3) is approximate."

Sentence: "The difference in observation angles 40° and 55°, respectively for SMAP and AMSR2, was neglected.» was delated from section 2. (The sentence in old version of the manuscript was between Line 95-100).

The list of references has been updated, see text highlighted in green in new version of manuscript.

For a more accurate understanding, proposed method, in the new version of the manuscript, we changed the notation in formula (3) for "$T_{s0}$" and provided links to additional literature.

Sentence

"For an isothermal and dielectric-homogeneous half-space, the reflectivity at frequency f, in accordance with the method (Muzalevskiy and Ruzicka, 2020), can be estimated based from equation (1) assuming $\frac{\Delta T_s}{\Delta z}\big|_{z=0} \equiv 0$:.."

was rewritten:

"For dielectric-inhomogeneous and nonisothermal half-space, the reflectivity at frequency f, in accordance with the method (Muzalevskiy et al., 2021; Muzalevskiy and Ruzicka, 2020), can be estimated based equation:

$$\Gamma_p(f) = 1 - \mathrm{Tb}_p(f)/\mathrm{T_{eff}}. \qquad (3)$$

In equation (3), $T_{eff}$ and $\Gamma_p(f)$ can be interpreted, respectively, as effective ground-canopy temperature and ground reflectivity, which takes into account soil surface roughness and canopy optical thickness using one combined parameter (Fernandez-Moran et al., 2015, see equation (10))."

Sentence

"The soil surface temperature $T_{s0}$ can be estimated based on the measurements of brightness temperature $\mathrm{Tb_V}(6.9)$ by the GCOM-W1 satellite on a frequency of 6.9 GHz in vertical polarization, the values of which correlate with the surface temperature of tundra soil (Muzalevskiy et al., 2016)."

was corrected

"The ==effective== temperature $T_{eff}$ can be estimated based on the measurements of brightness temperature $Tb_V(6.9)$ by the GCOM-W1 satellite on a frequency of 6.9 GHz in vertical polarization, the values of which correlate with the surface temperature of tundra soil (Muzalevskiy et al., 2016)."

**Comment 2.** Figure 1: Only Happy Valley test site is shown. I would argue its critical to somehow show the other test sites as well. At the very least there should be an example of one of the sites where the methodology does not work as well according to the authors.

**Response to comment 2.** We proceeded from the limited volume of the article. In accordance with your comment, we have added to Fig. 2 dependences $\Delta\Gamma_H(f)/\Delta f$ for test sites of KJ, CH, including SO, SA. In these areas, from our point of view, the method is quite suitable for use, the method reflects the objective situation of soil surface temperature fluctuations near 0C (please see response to comment 3 below). Contrary to the manuscript, below are figures for both $\Delta Tb_H(f)/\Delta f$ and $\Delta\Gamma_H/\Delta f$ for several test sites. Additional discussion of the new figures is provided in the response to comment 3 (see below).

[Figure]

[Figure]

Fig. 1. (Fig. 2 in manuscript). Test sites:(c) KJ, (d) CH, (e) SO, (f) SA, (g) FB

Caption to Fig. 2 in the new version of the manuscript was changed to:

" Figure 2. Density of spectral gradients of brightness temperature (a) and reflectivity (b) for HV test site, and density of spectral gradients of reflectivity for (c) KJ, (d) CH, (e) SO, (f) SA test sites, calculated for the pairs of frequencies: 1) 1.4-6.9 GHz, 2) 1.4-10.7 GHz, 3) 1.4-18.7 GHz and 4) 1.4-36.5 GHz."

**Comment 3.**

In section 2, the authors describe differences and similarities between the test sites. I can not see any discussion on how those conditions influence the outcome except for 'surface water area' which is briefly discussed but not with a focus on the differences between test sites. Please elaborate in the discussion section the soundness of the results for those different sites as it speaks to the transferability of your approach.

Line 181-182: Does this mean that this method is not applicable in lower latitudes/sub Arctic regions? Please expand/clarify.

**Response to comment 3.** In accordance with the comment, we had made further discussions. In the paragraph before Fig. 2 the following sentences have been added:

[revised manuscript text omitted]

Minor comments:

**Comment 4.** Line 134: ovals are not visible possibly replace with vertical line with a stronger color/stronger line width.

**Response to comment 4.** Fig.1a, 1b were replaced. In the new version of the manuscript the selected areas are shown as rectangles with a dash line. In the new version of the manuscript word "ovals" was replaced to "rectangles" on

Line 145: "Several such time periods are marked by rectangles with a dashed line in Fig. 1a, 1b (see the period 2015-2016). "

**Comment 5.** Figure 3: Figure caption to long, difficult to get the essential points from it. Please shorten. Also, it should be made clear in the figure caption that the different colors represent the different test sites.

**Response to comment 5.** Caption to Fig. 3 was replaced to

"Correlation between days of the year (DoY), for which: (a) $\Delta Tb_H/\Delta f$, (b) $\Delta\Gamma_H/\Delta f$, (c) MPR crosses threshold level, (d) FT soil state was detected (SMAP SPL3FTP_E product) and soil surface temperature stable crossing 0°C (weather stations data). Different test sites are marked with color symbols. The open and filled symbols indicate the frozen and thawed surface soil state, respectively. Squares and circles (regression lines) correspond to the spectral range of 6.9-1.4 GHz (1) and 36.5-1.4 GHz (2), respectively. Regression line (3) corresponds to MPR index and SMAP SPL3FTP_E product."

**Comment 6.** Line 188: swap identification with identify

**Response to comment 6.** Done.

**Comment 7.** Line 198: "FT topsoil state identification possible with determination coefficient": missing 'is'

**Response to comment 7.** Verb was inserted in sentence.

**Comment 8.** The word 'region' is used several times incorrectly in my opinion. Specifically, line 138 where it should be replaced by 'time period' or something similar.

**Response to comment 8.** It was corrected.

Sentence before Fig. 1 was corrected:

"Several such time periods are marked by rectangles with a dashed line in Fig. 1a, 1b (see the period 2015-2016)."

Sentence after Fig. 1 was corrected:

"In the transition period (May 7-19, 2016 and October-November, 2015, see Fig. 1)...."